# Synergistic HNO₃–H₂SO₄–NH₃ upper tropospheric particle formation

Mingyi Wang[1,2,7], Mao Xiao[3], Barbara Bertozzi[4], Guillaume Marie[5], Birte Rörup[6], Benjamin Schulze[7], Roman Bardakov[8,9], Xu-Cheng He[6], Jiali Shen[6], Wiebke Scholz[10], Ruby Marten[3], Lubna Dada[3,6], Rima Baalbaki[6], Brandon Lopez[1,11], Houssni Lamkaddam[3], Hanna E. Manninen[12], António Amorim[13], Farnoush Ataei[14], Pia Bogert[4], Zoé Brasseur[6], Lucía Caudillo[5], Louis-Philippe De Menezes[12], Jonathan Duplissy[6,15], Annica M. L. Ekman[8,9], Henning Finkenzeller[16], Loïc Gonzalez Carracedo[17], Manuel Granzin[5], Roberto Guida[12], Martin Heinritzi[5], Victoria Hofbauer[1,2], Kristina Höhler[4], Kimmo Korhonen[18], Jordan E. Krechmer[19], Andreas Kürten[5], Katrianne Lehtipalo[6,20], Naser G. A. Mahfouz[1,21], Vladimir Makhmutov[22,23], Dario Massabò[24], Serge Mathot[12], Roy L. Mauldin[1,2,25], Bernhard Mentler[10], Tatjana Müller[5,26], Antti Onnela[12], Tuukka Petäjä[6], Maxim Philippov[22], Ana A. Piedehierro[20], Andrea Pozzer[26], Ananth Ranjithkumar[27], Meredith Schervish[1,2], Siegfried Schobesberger[18], Mario Simon[5], Yuri Stozhkov[22], António Tomé[28], Nsikanabasi Silas Umo[4], Franziska Vogel[4], Robert Wagner[4], Dongyu S. Wang[3], Stefan K. Weber[12], André Welti[20], Yusheng Wu[6], Marcel Zauner-Wieczorek[5], Mikko Sipilä[6], Paul M. Winkler[17], Armin Hansel[10,29], Urs Baltensperger[3], Markku Kulmala[6,15,30,31], Richard C. Flagan[7], Joachim Curtius[5], Ilona Riipinen[9,32], Hamish Gordon[1,11], Jos Lelieveld[26,33], Imad El-Haddad[3], Rainer Volkamer[16], Douglas R. Worsnop[6,19], Theodoros Christoudias[33], Jasper Kirkby[5,12], Ottmar Möhler[4] & Neil M. Donahue[1,2,11,34 ✉]

New particle formation in the upper free troposphere is a major global source of cloud condensation nuclei (CCN)[1–4]. However, the precursor vapours that drive the process are not well understood. With experiments performed under upper tropospheric conditions in the CERN CLOUD chamber, we show that nitric acid, sulfuric acid and ammonia form particles synergistically, at rates that are orders of magnitude faster than those from any two of the three components. The importance of this mechanism depends on the availability of ammonia, which was previously thought to be efficiently scavenged by cloud droplets during convection. However, surprisingly high concentrations of ammonia and ammonium nitrate have recently been observed in the upper troposphere over the Asian monsoon region[5,6]. Once particles have formed, co-condensation of ammonia and abundant nitric acid alone is sufficient to drive rapid growth to CCN sizes with only trace sulfate. Moreover, our measurements show that these CCN are also highly efficient ice nucleating particles—comparable to desert dust. Our model simulations confirm that ammonia is efficiently convected aloft during the Asian monsoon, driving rapid, multi-acid HNO₃–H₂SO₄–NH₃ nucleation in the upper troposphere and producing ice nucleating particles that spread across the mid-latitude Northern Hemisphere.

Intense particle formation has been observed by airborne measurements as a persistent, global-scale band in the upper troposphere over tropical convective regions[1,2,4]. Upper tropospheric nucleation is thought to provide at least one-third of global CCN[3]. Increased aerosols since the industrial revolution, and their interactions with clouds, have masked a large fraction of the global radiative forcing by greenhouse gases. Projections of aerosol radiative forcing resulting from future reductions of air pollution are highly uncertain[7]. Present-day nucleation involves sulfuric acid (H₂SO₄) over almost all the troposphere[8]. However, binary nucleation of H₂SO₄–H₂O is slow and, so, ternary or multicomponent nucleation with extra vapours such as ammonia (NH₃)[9] and organics[10,11] is necessary to account for observed new-particle-formation rates[3,8,12].

Ammonia stabilizes acid–base nucleation and strongly enhances particle formation rates[9]. However, ammonia is thought to be extremely scarce in the upper troposphere because its solubility in water and reactivity with acids should lead to efficient removal in convective clouds. However, this assumption is not supported by observation. Ammonia vapour has been repeatedly detected in the Asian monsoon upper troposphere, with mixing ratios of up to 30 pptv (2.5 × 10⁸ cm⁻³) for a three-month average[5] and up to 1.4 ppbv (1.2 × 10¹⁰ cm⁻³) in hotspots[6].

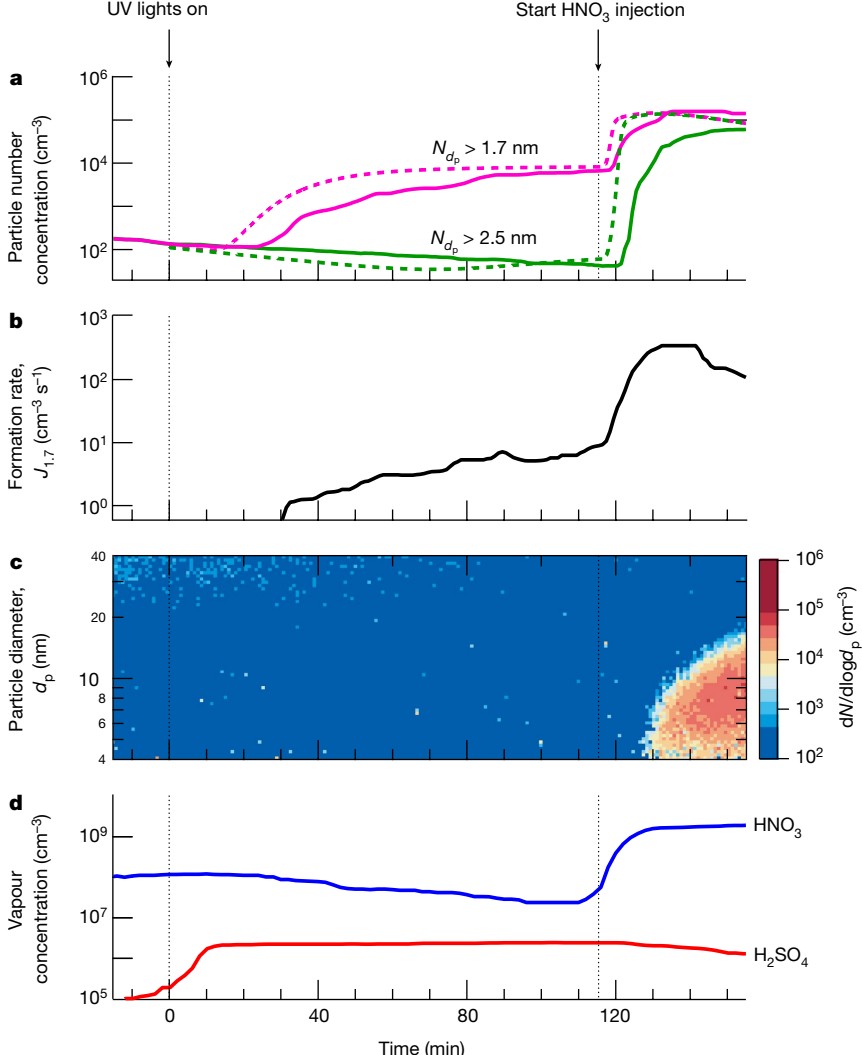

**Fig. 1 | Example experiment showing nitric acid enhancement of $H_2SO_4$–$NH_3$ particle formation. a,** Particle number concentrations versus time at mobility diameters >1.7 nm (magenta) and >2.5 nm (green). The solid magenta trace is measured by a $PSM_{1.7}$ and the solid green trace is measured by a $CPC_{2.5}$. The fixed experimental conditions are about $6.5 \times 10^8$ $cm^{-3}$ $NH_3$, 223 K and 25% relative humidity. A microphysical model reproduces the main features of the observed particle formation (dashed lines; see text for details). **b,** Particle formation rate versus time at 1.7 nm ($J_{1.7}$), measured by a PSM. **c,** Particle size distribution versus time, measured by an SMPS. **d,** Gas-phase nitric acid and sulfuric acid versus time, measured by an I⁻ CIMS and a $NO_3^-$ CIMS, respectively. Sulfuric acid through $SO_2$ oxidation started to appear soon after switching on the UV lights at time = 0 min, building up to a steady state of $2.3 \times 10^6$ $cm^{-3}$ after a wall-loss-rate timescale of around 10 min. The subsequent $H_2SO_4$–$NH_3$ nucleation led to a relatively slow formation rate of 1.7-nm particles. The particles did not grow above 2.5 nm because of their slow growth rate and corresponding low survival probability against wall loss. Following injection of $2.0 \times 10^9$ $cm^{-3}$ nitric acid into the chamber after 115 min, while leaving the production rate of sulfuric acid and the injection rate of ammonia unchanged, we observed a sharp increase in particle formation rate (panel **b**), together with rapid particle growth of 40 nm h⁻¹ (panel **c**). The overall systematic scale uncertainties of ±30% on particle formation rate, −33%/+50% on sulfuric acid concentration and ±25% on nitric acid concentration are not shown.

The release of dissolved ammonia from cloud droplets may occur during glaciation[13]. Once released in the upper troposphere, ammonia can form particles with nitric acid, which is abundantly produced by lightning[14,15]. These particles will live longer and travel farther than ammonia vapour, with the potential to influence the entire upper troposphere and lower stratosphere of the Northern Hemisphere[6].

Fundamental questions remain about the role and mechanisms of nitric acid and ammonia in upper tropospheric particle formation. Recent CLOUD (Cosmics Leaving Outdoor Droplets) experiments at CERN have shown that nitric acid and ammonia vapours below 278 K can condense onto newly formed particles as small as a few nanometres in diameter, driving rapid growth to CCN sizes[16]. At even lower temperatures (below 258 K), nitric acid and ammonia can directly nucleate to form ammonium nitrate particles, although pure $HNO_3$–$NH_3$ nucleation is too slow to compete with $H_2SO_4$–$NH_3$ nucleation under comparable conditions. However, the results we present here show that, when all three vapours are present, a synergistic interaction drives nucleation rates orders of magnitude faster than those from any two of the three components. Once nucleated through this multi-acid–ammonia mechanism, the particles can grow rapidly by co-condensation of $NH_3$ and $HNO_3$ alone, both of which may be far more abundant than $H_2SO_4$ in the upper troposphere.

## Particle formation measurements in CLOUD

Here we report new-particle-formation experiments performed with mixtures of sulfuric acid, nitric acid and ammonia vapours in the CLOUD chamber[9] at CERN between September and December 2019 (CLOUD 14; see Methods for experimental details). To span ranges typical of the upper troposphere, we established quasi-steady-state

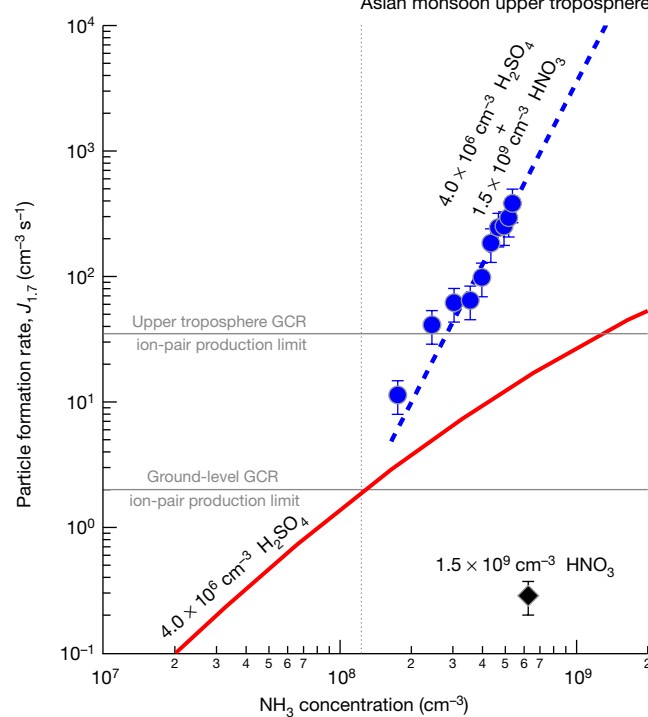

**Asian monsoon upper troposphere**

**Fig. 2 | Particle formation rates at 1.7 nm ($J_{1.7}$) versus ammonia concentration at 223 K and 25% relative humidity.** The chemical systems are $HNO_3$–$NH_3$ (black), $H_2SO_4$–$NH_3$ (red) and $HNO_3$–$H_2SO_4$–$NH_3$ (blue). The black diamond shows the CLOUD measurement of $HNO_3$–$NH_3$ nucleation at $1.5 \times 10^9$ cm$^{-3}$ $HNO_3$, $6.5 \times 10^8$ cm$^{-3}$ $NH_3$ and with $H_2SO_4$ below the detection limit of $5 \times 10^4$ cm$^{-3}$. The red solid curve is $J_{1.7}$ versus ammonia concentration at $4.0 \times 10^6$ cm$^{-3}$ sulfuric acid from a $H_2SO_4$–$NH_3$ nucleation parameterization on the basis of previous CLOUD measurements[18,19]. The blue circles show the CLOUD measurements of $HNO_3$–$H_2SO_4$–$NH_3$ nucleation at $4.0 \times 10^6$ cm$^{-3}$ $H_2SO_4$, $1.5 \times 10^9$ cm$^{-3}$ $HNO_3$ and $(1.6–6.5) \times 10^8$ cm$^{-3}$ $NH_3$. The data are fitted by a power law, $J_{1.7} = k[NH_3]^{3.7}$ (blue dashed curve). The vertical grey dotted line separates ammonia concentrations measured in different regions in the upper troposphere[5]; the region to the right indicates the Asian monsoon conditions. The horizontal grey solid lines show $J_{1.7}$ upper limits for ion-induced nucleation resulting from the GCR ionization rate of around 2 ion pairs cm$^{-3}$ s$^{-1}$ at ground level and 35 ion pairs cm$^{-3}$ s$^{-1}$ in the upper troposphere. Among the three nucleation mechanisms, $H_2SO_4$–$NH_3$ nucleation dominates in regions with low ammonia (below around $1.0 \times 10^8$ cm$^{-3}$, or 12 pptv), whereas $HNO_3$–$H_2SO_4$–$NH_3$ nucleation dominates at higher ammonia levels characteristic of the Asian monsoon upper troposphere. The bars indicate 30% estimated total error on the particle formation rates. The overall systematic scale uncertainties are −33%/+50% for sulfuric acid and ±25% for nitric acid concentrations.

vapour concentrations in the chamber of $(0.26–4.6) \times 10^6$ cm$^{-3}$ sulfuric acid (through photochemical oxidation of $SO_2$), $(0.23–4.0) \times 10^9$ cm$^{-3}$ nitric acid (through either photochemical oxidation of $NO_2$ or injection from an evaporator) and $(0.95–6.5) \times 10^8$ cm$^{-3}$ ammonia (through injection from a gas bottle). In an extreme experiment to simulate hotspot conditions in the Asian monsoon anticyclone, we raised sulfuric acid, nitric acid and ammonia to maximum concentrations of $6.2 \times 10^7$ cm$^{-3}$, $3.8 \times 10^9$ cm$^{-3}$ and $8.8 \times 10^9$ cm$^{-3}$, respectively. The experiments were conducted at 223 K and 25% relative humidity, representative of upper tropospheric conditions.

Figure 1 shows the evolution of a representative new-particle-formation experiment in the presence of around $6.5 \times 10^8$ cm$^{-3}$ ammonia. The top three panels show particle number concentrations above 1.7 nm and above 2.5 nm (Fig. 1a), particle formation rate at 1.7 nm ($J_{1.7}$) (Fig. 1b) and particle size distribution (Fig. 1c). The bottom panel shows $HNO_3$ and $H_2SO_4$ vapour concentrations (Fig. 1d). We switched on the

ultraviolet (UV) lights at $t = 0$ min to oxidize $SO_2$ with OH radicals and form $H_2SO_4$. Sulfuric acid started to appear shortly thereafter and built up to a steady state of $2.3 \times 10^6$ cm$^{-3}$ over the wall-loss timescale of about 10 min. Under these conditions, the data show a modest formation rate of 1.7-nm particles from $H_2SO_4$–$NH_3$ nucleation, consistent with previous CLOUD measurements[8]. These particles grew only slowly (about 0.5 nm h$^{-1}$ at this $H_2SO_4$ and particle size[17]). No particles reached 2.5 nm within 2 h, owing to their slow growth rate and low survival probability against wall loss.

At $t = 115$ min, we raised the nitric acid concentration to $2.0 \times 10^9$ cm$^{-3}$, through direct injection instead of photochemical production, so that we could independently control the nitric acid and sulfuric acid concentrations. The particle number increased 30-fold and 1,300-fold for particles larger than 1.7 nm and 2.5 nm, respectively. In addition, these newly formed particles grew much more rapidly (40 nm h$^{-1}$), reaching 20 nm within 30 min. This experiment shows that nitric acid can substantially enhance particle formation and growth rates for fixed levels of sulfuric acid and ammonia.

We also conducted model calculations on the basis of known thermodynamics and microphysics (Methods). Our model results (dashed traces in Fig. 1a) consistently and quantitatively confirm the experimental data: sulfuric acid and ammonia nucleation produces only 1.7-nm particles, whereas addition of nitric acid strongly enhances the formation rates of both 1.7-nm and 2.5-nm particles.

We conducted two further experiments under conditions similar to Fig. 1 but holding the concentrations of a different pair of vapours constant while varying the third. For the experiment shown in Extended Data Fig. 1, we started by oxidizing $NO_2$ to produce $1.6 \times 10^9$ cm$^{-3}$ $HNO_3$ in the presence of about $6.5 \times 10^8$ cm$^{-3}$ $NH_3$ and then increased $H_2SO_4$ from 0 to $4.9 \times 10^6$ cm$^{-3}$ by oxidizing progressively more injected $SO_2$. For the experiment shown in Extended Data Fig. 2, we first established $4.6 \times 10^6$ cm$^{-3}$ $H_2SO_4$ and $4.0 \times 10^9$ cm$^{-3}$ $HNO_3$, and then increased $NH_3$ from 0 to about $6.5 \times 10^8$ cm$^{-3}$. We consistently observed relatively slow nucleation when only two of the three vapours are present, whereas addition of the third vapour increased nucleation rates by several orders of magnitude.

Figure 2 shows particle formation rates measured by CLOUD at 1.7-nm mobility diameter ($J_{1.7}$) versus ammonia concentration, at 223 K. The $J_{1.7}$ data were all measured in the presence of ions from galactic cosmic rays (GCR) and − so − represent the sum of neutral and ion-induced channels. The black diamond shows the measured $J_{1.7}$ of 0.3 cm$^{-3}$ s$^{-1}$ for $HNO_3$–$NH_3$ nucleation with $1.5 \times 10^9$ cm$^{-3}$ nitric acid, about $6.5 \times 10^8$ cm$^{-3}$ ammonia and sulfuric acid below the detection limit of $5 \times 10^4$ cm$^{-3}$ (this is the event shown in Extended Data Fig. 1). At this same ammonia concentration, we measured $J_{1.7} = 6.1$ cm$^{-3}$ s$^{-1}$ at $2.3 \times 10^6$ cm$^{-3}$ $H_2SO_4$, demonstrating the much faster rate of $H_2SO_4$–$NH_3$ nucleation (not shown). This measurement is consistent with models on the basis of previous CLOUD studies of $H_2SO_4$–$NH_3$ nucleation[18,19], as illustrated by the model simulations for $4.0 \times 10^6$ cm$^{-3}$ sulfuric acid (red solid curve). The blue circles show our measurements of $J_{1.7}$ for $HNO_3$–$H_2SO_4$–$NH_3$ nucleation at $4.0 \times 10^6$ cm$^{-3}$ sulfuric acid and $(1.6–6.5) \times 10^8$ cm$^{-3}$ ammonia, in the presence of $1.5 \times 10^9$ cm$^{-3}$ nitric acid (the event shown in Extended Data Fig. 2). The blue dashed curve is a power law fit to the measurements, indicating a strong sensitivity to ammonia concentration ($J_{1.7} = k[NH_3]^{3.7}$).

The vertical grey dotted line in Fig. 2 separates ammonia concentrations measured in different regions in the upper troposphere[5]; Asian monsoon conditions are to the right of this vertical line. Our results indicate that $H_2SO_4$–$NH_3$ nucleation is probably responsible for new particle formation in regions with ammonia concentrations below around $10^8$ cm$^{-3}$ (12 pptv), but that $HNO_3$–$H_2SO_4$–$NH_3$ nucleation probably dominates at higher ammonia levels in the Asian monsoon upper troposphere. Our nucleation rate measurements confirm that the stronger sulfuric acid is favoured by ammonia in the ammonia-limited regime, so nitric acid will evaporate from the clusters, as it may be displaced by

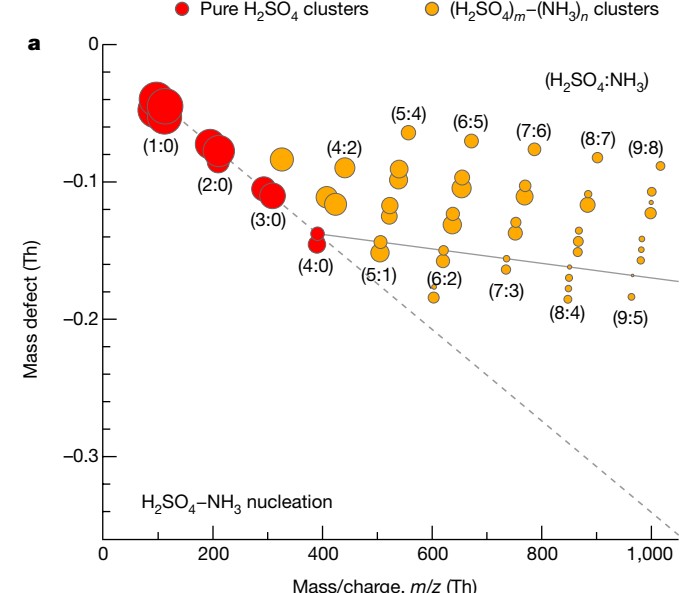

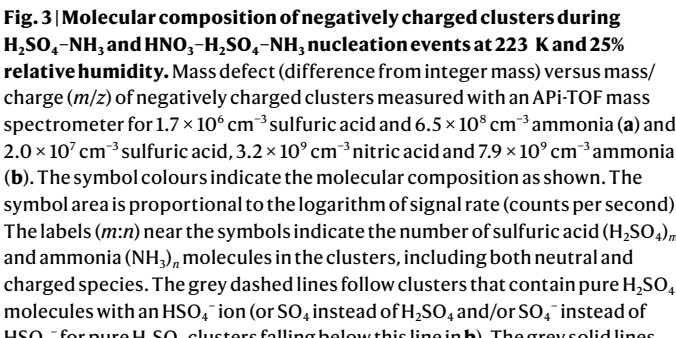

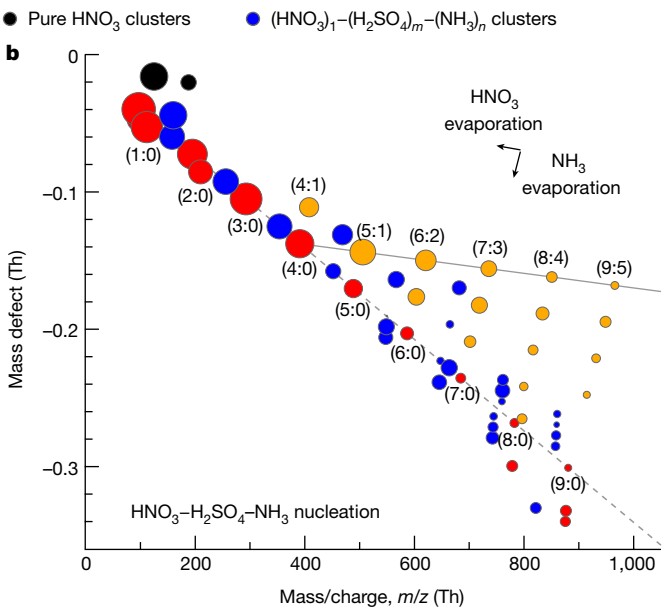

**Fig. 3 | Molecular composition of negatively charged clusters during $H_2SO_4$–$NH_3$ and $HNO_3$–$H_2SO_4$–$NH_3$ nucleation events at 223 K and 25% relative humidity.** Mass defect (difference from integer mass) versus mass/charge ($m/z$) of negatively charged clusters measured with an APi-TOF mass spectrometer for $1.7 \times 10^6$ cm$^{-3}$ sulfuric acid and $6.5 \times 10^8$ cm$^{-3}$ ammonia (**a**) and $2.0 \times 10^7$ cm$^{-3}$ sulfuric acid, $3.2 \times 10^9$ cm$^{-3}$ nitric acid and $7.9 \times 10^9$ cm$^{-3}$ ammonia (**b**). The symbol colours indicate the molecular composition as shown. The symbol area is proportional to the logarithm of signal rate (counts per second). The labels ($m$:$n$) near the symbols indicate the number of sulfuric acid ($H_2SO_4$)$_m$ and ammonia ($NH_3$)$_n$ molecules in the clusters, including both neutral and charged species. The grey dashed lines follow clusters that contain pure $H_2SO_4$ molecules with an $HSO_4^-$ ion (or $SO_4$ instead of $H_2SO_4$ and/or $SO_4^-$ instead of $HSO_4^-$ for pure $H_2SO_4$ clusters falling below this line in **b**). The grey solid lines

follow the 1:1 $H_2SO_4$–$NH_3$ addition starting at $(H_2SO_4)_4$–$(NH_3)_0$. Nearly all clusters in panel **a** lie above this line, whereas nearly all clusters in panel **b** fall below it. Most clusters containing $HNO_3$ lack $NH_3$ by the time they are measured (they fall near the ($m$:0) grey dashed line), but the marked difference between **a** and **b** indicates that the nucleating clusters had distinctly different compositions, probably including relatively weakly bound $HNO_3$–$NH_3$ pairs in **b**. It is probable that nucleating clusters in the CLOUD chamber at 223 K contain $HNO_3$–$H_2SO_4$–$NH_3$ with a roughly 1:1 acid–base ratio. However, during the transmission from the chamber to the warm APi-TOF mass spectrometer at 293 K, the clusters lose $HNO_3$ and $NH_3$, leaving a less volatile core of $H_2SO_4$ with depleted $NH_3$. The evaporation of a single $NH_3$ or $HNO_3$ molecule from a cluster displaces it on the mass defect plot by a vector distance indicated by the black arrows in **b**.

sulfuric acid. However, as ammonia increases from 1.6 to $6.5 \times 10^8$ cm$^{-3}$, we observe sharp increases in $J_{1.7}$ for $HNO_3$–$H_2SO_4$–$NH_3$ nucleation from 10 to 400 cm$^{-3}$ s$^{-1}$ and in the ratio of particle formation rates ($HNO_3$–$H_2SO_4$–$NH_3$:$H_2SO_4$–$NH_3$) from 4 to 30. Our nucleation model (as in Fig. 1) yields slightly higher $J_{1.7}$ than that observed, as shown in Extended Data Fig. 3, but the formation rate variation with ammonia, nonetheless, shows a similar slope.

CLOUD has previously shown that ions enhance nucleation for all but the strongest acid–base clusters; $HNO_3$–$H_2SO_4$–$NH_3$ is probably not an exception. However, the ion enhancement is limited by the GCR ion-pair production rate. We show with the horizontal grey solid lines in Fig. 2 the upper limits on $J_{1.7}$ for ion-induced nucleation of about 2 cm$^{-3}$ s$^{-1}$ at ground level and 35 cm$^{-3}$ s$^{-1}$ in the upper troposphere. Our experimental nucleation rates for $HNO_3$–$H_2SO_4$–$NH_3$ are mostly above upper tropospheric GCR ion production rates. This is confirmed by similar $J_{1.7}$ measured during a neutral nucleation experiment, in which an electric field was used to rapidly sweep ions from the chamber. Thus, for this nucleation scheme, the neutral channel will often prevail over the ion-induced channel in the Asian monsoon upper troposphere. However, when ammonia is diluted away outside the Asian monsoon anticyclone, ions may enhance the nucleation rate up to the GCR limit near 35 cm$^{-3}$ s$^{-1}$.

In a formal sense, the new-particle-formation mechanism could be one of two types: formation of stable $H_2SO_4$–$NH_3$ clusters, followed by nano-Köhler-type activation by nitric acid and ammonia[16]; or else true synergistic nucleation of nitric acid, sulfuric acid and ammonia[9]. In a practical sense, it makes little difference because coagulation loss is a major sink for all small clusters in the atmosphere[20], so appearance of 1.7-nm particles by means of any mechanism constitutes new particle

formation. Regardless, we can distinguish between these two possibilities from our measurements of the molecular composition of negatively charged clusters using an atmospheric pressure interface time-of-flight (APi-TOF) mass spectrometer. In Fig. 3, we show cluster mass defect plots during $H_2SO_4$–$NH_3$ and $HNO_3$–$H_2SO_4$–$NH_3$ nucleation events at 223 K. The marked difference between Fig. 3a, b indicates that nitric acid changes the composition of the nucleating clusters down to the smallest sizes; thus, the mechanism is almost certainly synergistic $HNO_3$–$H_2SO_4$–$NH_3$ nucleation.

In Fig. 3a, the predominant ions are one of several deprotonated sulfuric acid species, including $HSO_4^-$, $SO_4^-$, $HSO_5^-$, $SO_5^-$ and so on, resulting in a group of points for clusters with similar molecular composition but different mass and mass defect. In the figure, we use the labels ($m$:$n$) to indicate the number of sulfuric acid and ammonia molecules in the $(H_2SO_4)_m$–$(NH_3)_n$ clusters, including both neutral and charged species. The mass defect plot closely resembles those previously measured for $H_2SO_4$–$NH_3$ nucleation[21]. Negative-ion-induced nucleation proceeds with the known acid–base stabilization mechanism, in which sulfuric acid dimers form as a first step (with $HSO_4^-$ serving as a conjugate base for the first $H_2SO_4$) and then clusters subsequently grow by 1:1 $H_2SO_4$–$NH_3$ addition (that is, as ammonium bisulfate)[9]. We use a grey line to illustrate the 1:1 addition path, beginning at $(H_2SO_4)_4$–$(NH_3)_0$. Clusters larger than the sulfuric acid tetramers mostly contain several ammonia molecules and, so nearly all clusters in Fig. 3a lie above the grey line.

Figure 3b shows a pronounced change in the cluster APi-TOF signal during $HNO_3$–$H_2SO_4$–$NH_3$ nucleation. In addition to pure $(H_2SO_4)_m$–$(NH_3)_n$ clusters, we observe clusters with one extra $HNO_3$ molecule (or $NO_3^-$ ion), that is, $(HNO_3)_1$–$(H_2SO_4)_m$–$(NH_3)_n$, and the pure nitric acid

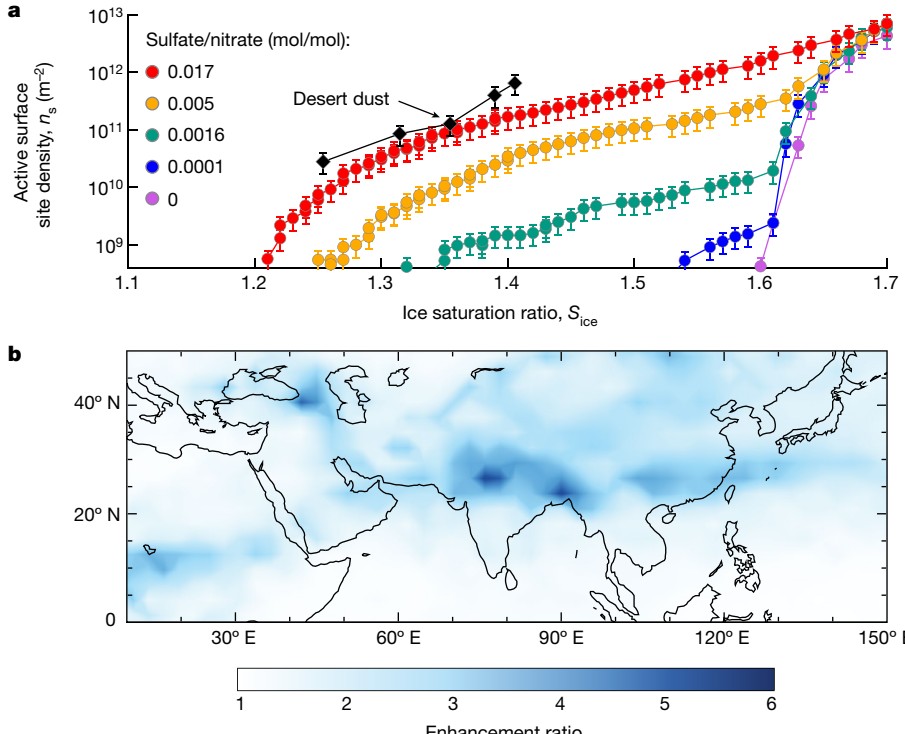

**Fig. 4 | Ice nucleation properties and modelled regional contribution of upper tropospheric particles formed from HNO₃–H₂SO₄–NH₃ nucleation.** a, Active surface site density versus ice saturation ratio, measured by the mINKA instrument at CLOUD, at 233 K and 25% relative humidity. Pure ammonium nitrate particles (purple points) show homogeneous freezing. However, addition of only small amounts of sulfate creates highly ice-nucleation-active particles. At around 1.7% sulfate fraction (red points), the ice nucleating efficiency is comparable with desert dust particles[24].

b, Simulation of particle formation in a global model (EMAC) with efficient vertical transport of ammonia into the upper troposphere during the Asian monsoon. Including multi-acid HNO₃–H₂SO₄–NH₃ nucleation (on the basis of the blue dashed curve in Fig. 2) enhances particle number concentrations (nucleation mode) over the Asian monsoon region by a factor of 3–5 compared with the same model with only H₂SO₄–NH₃ nucleation (from Dunne et al.[8], similar to the red solid curve in Fig. 2).

monomer and dimer. In sharp contrast with Fig. 3a, all these clusters are deficient in NH₃, falling below the same grey line as in Fig. 3a. The most deficient contain up to nine bare acids, that is, $(H_2SO_4)_9$ or $(H_2SO_4)_8$–$(HNO_3)_1$. Figure 3b almost certainly does not represent the true cluster composition in the chamber because binary nucleation of $H_2SO_4$ does not proceed under these exact conditions of $H_2SO_4$, $NH_3$, temperature and relative humidity (as demonstrated by Fig. 3a). We can interpret Fig. 3b as follows. It is probable that clusters in the CLOUD chamber (223 K) contain HNO₃–H₂SO₄–NH₃ with a roughly 1:1 acid–base ratio, representing partial neutralization. However, during the transmission from the cold chamber to the warm APi-TOF mass spectrometer (about 293 K), the clusters lose relatively weakly bound HNO₃ and NH₃ molecules but not the lower-volatility $H_2SO_4$ molecules. Regardless of the interpretation, however, the notable difference between Fig. 3a, b indicates that the sampled clusters had very different compositions and that nitric acid participated in the formation of clusters as small as a few molecules.

## Ice nucleation measurements

Nitric acid and ammonia not only enhance the formation rate of new particles but also drive their rapid growth to sizes at which they may act as CCN or ice nucleating particles (INP), above around 50 nm. To assess their effect on cirrus clouds, we measured the ice nucleation ability of particles formed from HNO₃–H₂SO₄–NH₃ nucleation in the CLOUD chamber. Simulating 'hotspot' conditions, we first formed pure ammonium nitrate particles by means of HNO₃–NH₃ nucleation and then increased the $H_2SO_4$ fraction in the particles by oxidizing progressively more $SO_2$. We measured their ice nucleation ability using

the online continuous flow diffusion instrument, mINKA (Methods and Extended Data Fig. 4). As shown in Fig. 4a, pure ammonium nitrate particles (purple data points) nucleate ice only at high ice saturation ratios ($S_{ice}$), characteristic of homogeneous nucleation (shown by a steep increase of ice activation above $S_{ice} = 1.60$ at 215 K). This indicates that pure ammonium nitrate particles, formed by means of HNO₃–NH₃ nucleation, are probably in a liquid state initially, albeit at a relative humidity below the deliquescence point[22]. However, addition of sulfate, with a particulate sulfate-to-nitrate molar ratio as small as $10^{-4}$, triggers crystallization of ammonium nitrate. For these particles, we observed a small heterogeneous ice nucleation mode at $S_{ice}$ of 1.54 (blue data points), with other conditions and the particle size distribution held almost constant. Moreover, as the sulfate molar fraction progressively rises to just 0.017 (still almost pure but now solid ammonium nitrate), an active surface site density ($n_s$) of $10^{10}$ m⁻² is reached at $S_{ice}$ as low as 1.26. This is consistent with previous findings, in which particles were generated through nebulization, with a much larger particle diameter and a much higher sulfate-to-nitrate ratio[23]. Our measurements show that HNO₃–H₂SO₄–NH₃ nucleation followed by rapid growth from nitric acid and ammonia condensation – which results in low sulfate-to-nitrate ratio – could provide an important source of INP that are comparable with typical desert dust particles at nucleating ice[24].

## Atmospheric implications

Our findings suggest that HNO₃–H₂SO₄–NH₃ nucleation may dominate new particle formation in the Asian monsoon region of the upper troposphere, with a 'flame' of new particles in the outflow of convective

clouds, in which up to $10^{10}$ cm$^{-3}$ ammonia[6] mixes with low (background) levels of sulfuric acid and nitric acid. Without this mechanism, particle formation through the traditional ternary $H_2SO_4$–$NH_3$ nucleation would be much slower and most probably rate-limited by the scarce sulfuric acid. Furthermore, by co-condensing with nitric acid, the convected ammonia also drives the growth of the newly formed particles. Given typical acid-excess conditions in the upper troposphere, condensational growth is governed by the availability of ammonia. Consequently, particles will steadily (and rapidly) grow until ammonia is depleted after several *e*-folding times set by the particle condensation sink. On the basis of condensation sinks generally observed in the tropical upper troposphere[4], this timescale will be several hours. Within this time interval, given the observed ammonia levels, newly formed particles will be able to grow to CCN sizes and even small admixtures of sulfuric acid will render these particles efficient INP.

Our laboratory measurements provide a mechanism that can account for recent observations of abundant ammonium nitrate particles in the Asian monsoon upper troposphere[6]. To evaluate its importance on a global scale, we first parameterized our experimentally measured $J_{1.7}$ for $HNO_3$–$H_2SO_4$–$NH_3$ nucleation as a function of sulfuric acid, nitric acid and ammonia concentrations (Methods). The parameterization is obtained using a power-law dependency for each vapour (Extended Data Fig. 5), given that the critical cluster composition is associated with the exponents according to the first nucleation theorem[25]. Then we implemented this parameterization in a global aerosol model (EMAC, see Methods for modelling details). The EMAC model predicts that $HNO_3$–$H_2SO_4$–$NH_3$ nucleation at 250 hPa (11 km, approximately 223 K) produces an annual average exceeding 1,000 cm$^{-3}$ new particles over an extensive area (Extended Data Fig. 6). This corresponds to an increase in particle number concentration (Fig. 4b) up to a factor of five higher than in a control simulation with only ternary $H_2SO_4$–$NH_3$ nucleation[8]. The strongest increase occurs mostly over Asia, in which ammonia is ample because of deep convection from ground sources.

However, another global model (TOMCAT, see Methods) shows much lower ammonia mixing ratios in the upper troposphere than EMAC (<1 pptv compared with <100 pptv, respectively), although with a broadly similar spatial distribution (Extended Data Fig. 7a, b). This large variability of upper tropospheric ammonia is also indicated by recent field measurements on local[6,26] and global[5,27] scales. In view of its importance for both $H_2SO_4$–$NH_3$ and $HNO_3$–$H_2SO_4$–$NH_3$ nucleation, there is an urgent need to improve upper tropospheric measurements of ammonia, as well as improve knowledge of its sources, transport and sinks.

We thus turned to a cloud-resolving model to estimate the ammonia vapour fraction remaining after deep convection (see Methods). We show in Extended Data Fig. 8 that around 10% of the boundary layer ammonia can be transported into the upper troposphere and released as vapour by a base-case convective cloud. The sensitivity tests further illustrate that the key factor governing the fraction of ammonia remaining in the cloud outflow is the retention of ammonia molecules by ice particles (Extended Data Fig. 8e), whereas cloud water pH (Extended Data Fig. 8c) and cloud water content (Extended Data Fig. 8d) only play minor roles once glaciation occurs. Given that more than 10 ppbv of ammonia is often observed in the Asian boundary layer[28], it is plausible that the observed 1.4 ppbv ($10^{10}$ cm$^{-3}$) ammonia in the upper troposphere[6] is indeed efficiently transported by the convective systems.

Although the ammonium–nitrate–sulfate particles are formed locally, they can travel from Asia to North America in just three days by means of the subtropical jet stream, as the typical residence time of Aitken mode particles ranges from one week to one month in the upper troposphere[29]. As a result, these particles can persist as an intercontinental band, covering more than half of the mid-latitude surface area of the Northern Hemisphere (Extended Data Fig. 6). In summary, synergistic nucleation of nitric acid, sulfuric acid and ammonia could provide an important source of new CCN and ice nuclei in the upper troposphere, especially over the Asian monsoon region, and is closely linked with anthropogenic ammonia emissions[27].

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

[1]Center for Atmospheric Particle Studies, Carnegie Mellon University, Pittsburgh, PA, USA. [2]Department of Chemistry, Carnegie Mellon University, Pittsburgh, PA, USA. [3]Laboratory of Atmospheric Chemistry, Paul Scherrer Institute, Villigen, Switzerland. [4]Institute of Meteorology and Climate Research, Karlsruhe Institute of Technology, Karlsruhe, Germany. [5]Institute for Atmospheric and Environmental Sciences, Goethe University Frankfurt, Frankfurt am Main, Germany. [6]Institute for Atmospheric and Earth System Research (INAR), University of Helsinki, Helsinki, Finland. [7]Present address: Division of Chemistry and Chemical Engineering, California Institute of Technology, Pasadena, CA, USA. [8]Department of Meteorology, Stockholm University, Stockholm, Sweden. [9]Bolin Centre for Climate Research, Stockholm University, Stockholm, Sweden. [10]Institute for Ion Physics and Applied Physics, University of Innsbruck, Innsbruck, Austria. [11]Department of Chemical Engineering, Carnegie Mellon University, Pittsburgh, PA, USA. [12]CERN, the European Organization for Nuclear Research, Geneva, Switzerland. [13]CENTRA and Faculdade de Ciências da Universidade de Lisboa, Campo Grande, Lisbon, Portugal. [14]Leibniz Institute for Tropospheric Research, Leipzig, Germany. [15]Helsinki Institute of Physics, University of Helsinki, Helsinki, Finland. [16]Department of Chemistry & CIRES, University of Colorado Boulder, Boulder, CO, USA. [17]Faculty of Physics, University of Vienna, Vienna, Austria. [18]Department of Applied Physics, University of Eastern Finland, Kuopio, Finland. [19]Aerodyne Research, Inc., Billerica, MA, USA. [20]Finnish Meteorological Institute, Helsinki, Finland. [21]Atmospheric and Oceanic Sciences, Princeton University, Princeton, NJ, USA. [22]P. N. Lebedev Physical Institute of the Russian Academy of Sciences, Moscow, Russia. [23]Moscow Institute of Physics and Technology (National Research University), Moscow, Russia. [24]Department of Physics, University of Genoa & INFN, Genoa, Italy. [25]Department of Atmospheric and Oceanic Sciences, University of Colorado Boulder, Boulder, CO, USA. [26]Atmospheric Chemistry Department, Max Planck Institute for Chemistry, Mainz, Germany. [27]School of Earth and Environment, University of Leeds, Leeds, UK. [28]Institute Infante Dom Luíz, University of Beira Interior, Covilhã, Portugal. [29]Ionicon Analytik Ges.m.b.H., Innsbruck, Austria. [30]Joint International Research Laboratory of Atmospheric and Earth System Sciences, Nanjing University, Nanjing, China. [31]Aerosol and Haze Laboratory, Beijing Advanced Innovation Center for Soft Matter Science and Engineering, Beijing University of Chemical Technology, Beijing, China. [32]Department of Environmental Science (ACES), Stockholm University, Stockholm, Sweden. [33]Climate and Atmosphere Research Center, The Cyprus Institute, Nicosia, Cyprus. [34]Department of Engineering and Public Policy, Carnegie Mellon University, Pittsburgh, PA, USA. ✉e-mail: nmd@andrew.cmu.edu

## Methods

### The CLOUD facility

We conducted our measurements at the CERN CLOUD facility, a 26.1-m$^3$, electropolished, stainless-steel CLOUD chamber that allows new-particle-formation experiments under the full range of tropospheric conditions with scrupulous cleanliness and minimal contamination[9,30]. The CLOUD chamber is mounted in a thermal housing, capable of keeping the temperature constant in the range 208 K and 373 K with a precision of ±0.1 K (ref. [31]). Photochemical processes are initiated by homogeneous illumination with a built-in UV fibre-optic system, including four 200-W Hamamatsu Hg-Xe lamps at wavelengths between 250 and 450 nm and a 4-W KrF excimer UV laser at 248 nm with adjustable power. New particle formation under different ionization levels is simulated with and without the electric fields (±30 kV), which can artificially scavenge or preserve small ions produced from ground-level GCR. Uniform spatial mixing is achieved with magnetically coupled stainless-steel fans mounted at the top and bottom of the chamber. The characteristic gas mixing time in the chamber during experiments is a few minutes. The loss rate of condensable vapours and particles onto the chamber walls is comparable with the ambient condensation sink. To avoid contamination, the chamber is periodically cleaned by rinsing the walls with ultra-pure water and heating to 373 K for at least 24 h, ensuring extremely low contaminant levels of sulfuric acid $<5 \times 10^4$ cm$^{-3}$ and total organics <50 pptv (refs. [32,33]). The CLOUD gas system is also built to the highest technical standards of cleanliness and performance. The dry air supply for the chamber is provided by boil-off oxygen (Messer, 99.999%) and boil-off nitrogen (Messer, 99.999%) mixed at the atmospheric ratio of 79:21. Highly pure water vapour, ozone and other trace gases such as nitric acid and ammonia can be precisely added at the pptv level from ultra-pure sources.

### Instrumentation

Gas-phase sulfuric acid was measured using a nitrate chemical ionization APi-TOF (nitrate-CI-APi-TOF) mass spectrometer[34,35] and an iodide chemical ionization time-of-flight mass spectrometer equipped with a Filter Inlet for Gases and Aerosols (I-FIGAERO-CIMS)[36,37]. The nitrate-CI-APi-TOF mass spectrometer is equipped with an electrostatic filter in front of the inlet to remove ions and charged clusters formed in the chamber. A corona charger is used to ionize the reagent nitric acid vapour in a nitrogen flow[38]. Nitrate ions are then guided in an atmospheric pressure drift tube by an electric field to react with the analyte molecules in the sample flow. Sulfuric acid is quantified for the nitrate-CI-APi-TOF with a detection limit of about $5 \times 10^4$ cm$^{-3}$, following the same calibration and loss correction procedures described previously[9,32,39]. FIGAERO is a manifold inlet for a CIMS with two operating modes. In the sampling mode, a coaxial core sampling is used to minimize the vapour wall loss in the sampling line. The total flow is maintained at 18.0 slpm and the core flow at 4.5 slpm; the CIMS samples at the centre of the core flow with a flow rate of 1.6 slpm. Analyte molecules are introduced into a 150-mbar ion-molecule reactor, chemically ionized by iodide ions that are formed in a Po-210 radioactive source and extracted into the mass spectrometer. The sulfuric acid calibration coefficient for the I-FIGAERO-CIMS is derived using the absolute sulfuric acid concentrations measured with the pre-calibrated nitrate-CI-APi-TOF.

Gas-phase nitric acid was also measured using the I-FIGAERO-CIMS. Nitric acid concentration was quantified by measuring HNO$_3$/N$_2$ mixtures with known nitric acid concentrations, following similar procedures described previously[16]. The HNO$_3$/N$_2$ mixture was sourced from flowing 2 slpm ultra-pure nitrogen through a portable nitric acid permeation tube, at constant 40 °C. The permeation rate of nitric acid was determined by passing the outflow of the permeation tube through an impinger containing deionized water and analysing the resulting nitric acid solution through spectrophotometry.

Gas-phase ammonia was either measured or calculated. We measured ammonia using a proton transfer reaction time-of-flight mass spectrometer (PTR3-TOF-MS, or PTR3 for short)[40]. As a carrier gas for the primary ions, we used argon (ultra-high purity 5.0) to ensure that ammonium ions could not be artificially formed in the region of the corona discharge. Although the theoretical detection limit from peak height and width would be even smaller, the lowest concentration we were able to measure during the first fully ammonia-free runs of the beginning of the campaign was 10$^9$ cm$^{-3}$. An explanation for this is that, when concentrations of ammonia are low, effects of wall interaction of the highly soluble ammonia become important and the decay of ammonia in the inlet line becomes very slow. To reduce inlet wall contacts, we used a core-sampling technique directly in front of the instrument to sample only the centre 2 slpm of the 10 slpm inlet flow, but owing to frequent necessary on-site calibrations of volatile organic compounds, a Teflon ball valve was placed within the sample line that probably influenced measurements during times of low ammonia concentrations. At concentrations above about $2 \times 10^9$ cm$^{-3}$ ammonia, however, the response of the instrument was very fast, so that, for example, changes in the chamber ammonia flow rate were easily detectable. Off-site calibrations showed a humidity-independent calibration factor of 0.0017 ncps/ppb. Calibrated data from the PTR3 agree very well with the Picarro above 10$^{10}$ cm$^{-3}$ (detection limit of the Picarro). The PTR3 also provides information about the overall cleanliness of the volatile organic compounds in the chamber. The technique was extensively described previously[40].

For ammonia concentrations below 10$^9$ cm$^{-3}$, we calculated concentration using the calibrated ammonia injection flow and an estimated first-order wall-loss rate. The wall-loss rate ($k_{wall}$) for ammonia inside the CLOUD chamber is confirmed to be faster than for sulfuric acid[41], and can be determined from the following expression[42]:

$$k_{wall} = \frac{A}{V} \frac{2}{\pi} \sqrt{k_e\, D_i} = C_{wall} \sqrt{D_i} \qquad (1)$$

in which $A/V$ is the surface-to-volume ratio of the chamber, $k_e$ is the eddy diffusion constant (determined by the turbulent mixing intensity, not the transport properties of the gases) and $D_i$ is the diffusion coefficient for each gas. $C_{wall}$ is thus referred to as an empirical parameter of experiment conditions in the chamber. Here we first determine the $k_{wall}$ for sulfuric acid and nitric acid to be $1.7 \times 10^{-3}$ and $1.9 \times 10^{-3}$ s$^{-3}$, respectively, by measuring their passive decay rates and subtracting the loss rate of chamber dilution for both ($1.2 \times 10^{-3}$ s$^{-1}$), as well as the loss rate of dimer formation for sulfuric acid (around $1.6 \times 10^{-3}$ s$^{-1}$ for $5 \times 10^6$ cm$^{-3}$ H$_2$SO$_4$). The $k_{wall}$ for sulfuric acid agrees with our measurements from previous campaigns[43]. We then derive the $C_{wall}$ for sulfuric acid and nitric acid both to be $2.0 \times 10^{-4}$ torr$^{-0.5}$ cm$^{-1}$ s$^{-0.5}$, with $D_{H_2SO_4}$ of 74 torr cm$^2$ s$^{-1}$ and $D_{HNO_3}$ of 87 torr cm$^2$ s$^{-1}$ (ref. [44]). Finally, we calculate the $k_{wall}$ for ammonia to be $2.7 \times 10^{-3}$ s$^{-1}$, with $D_{NH_3}$ of 176 torr cm$^2$ s$^{-1}$ (ref. [44]). Ammonia desorption from the chamber surface is a strong function of the temperature and is believed to be negligible at low temperatures[30]. Even after a long time exposure, ammonia desorption should be less than $1.6 \times 10^6$ cm$^{-3}$, according to previous parameterization of ammonia background contamination in the CLOUD chamber[41].

The composition of negatively charged ions and clusters were determined using an APi-TOF mass spectrometer[45]. The APi-TOF mass spectrometer is connected to the CLOUD chamber by means of a 1-inch (21.7-mm inner diameter) sampling probe, with coaxial core sampling to minimize the wall losses in the sampling line. The total sample flow is maintained at 20 slpm and the core sample flow for the APi-TOF mass spectrometer at 0.8 slpm. Because this instrument only measures charged clusters, the measurements were made during GCR conditions. Owing to a large temperature difference between the cold chamber (223 K) and the warm APi-TOF mass spectrometer (around 293 K), HNO$_3$–H$_2$SO$_4$–NH$_3$ clusters probably lose relatively

weakly bonded $HNO_3$ and $NH_3$ molecules. This resembles the chemical ionization process of detecting ammonia with the nitrate-CI-APi-TOF, in which $HNO_3$ and $NH_3$ molecules rapidly evaporate from the resulting ammonia nitrate cluster in the CI-APi-TOF vacuum regions[46].

Gas monitors were used to measure ozone ($O_3$, Thermo Environmental Instruments TEI 49C), sulfur dioxide ($SO_2$, Thermo Fisher Scientific Inc. 42i-TLE) and nitric oxide (NO, ECO Physics, CLD 780TR). Nitrogen dioxide ($NO_2$) was measured by a cavity attenuated phase shift nitrogen dioxide monitor (CAPS $NO_2$, Aerodyne Research Inc.) and a home-made cavity enhanced differential optical absorption spectroscopy (CE-DOAS) instrument. The relative humidity of the chamber was determined by dew point mirrors (EdgeTech).

Particle number concentrations were monitored by condensation particle counters (CPCs), including an Airmodus A11 nano Condensation Nucleus Counter (nCNC), consisting of a particle size magnifier (PSM) and a laminar-flow butanol-based CPC[47], as well as a butanol TSI 3776 CPC. Particle size distributions between 1.8 nm and 500 nm were measured by a nano-scanning electrical mobility spectrometer (nSEMS), a nano-scanning mobility particle sizer (nano-SMPS) and a long-SMPS. The nSEMS used a new, radial opposed migration ion and aerosol classifier (ROMIAC), which is less sensitive to diffusional resolution degradation than the DMAs[48], and a soft X-ray charge conditioner. After leaving the classifier, particles were first activated in a fast-mixing diethylene glycol stage[49] and then counted with a butanol-based CPC. The nSEMS transfer function that was used to invert the data to obtain the particle size distribution was derived using 3D finite element modelling of the flows, electric field and particle trajectories[50,51]. The two commercial mobility particle size spectrometers, nano-SMPS and long-SMPS, have been fully characterized, calibrated and validated in several previous studies[52–54].

Particle-phase chemical composition was quantified using a high-resolution time-of-flight aerosol mass spectrometer (HR-ToF-AMS, Aerodyne Research). The working principles of the HR-ToF-AMS have been explained in detail previously[55,56]. In brief, particles are focused by an aerodynamic lens and flash-vaporized by impact onto a hot surface at 600 °C under a high vacuum. The vapours are then ionized by 70-eV electrons and the ions are detected with a ToF mass spectrometer. Ionization efficiency calibrations were conducted before and after the campaign and the variation is within 30%. The particle collection efficiency was considered constant during the experiments because temperature and relative humidity in the chamber were fixed and the particle composition was dominated by ammonium nitrate.

INP were measured in real time at 215 K, as a function of ice saturation ratio ($S_{ice}$), by the mobile ice nucleation instrument of the Karlsruhe Institute of Technology (mINKA). mINKA is a continuous flow diffusion chamber with vertical cylindrical geometry[57], on the basis of the design of INKA[58,59]. A detailed description of the continuous flow diffusion chamber working principle is presented elsewhere[57]. Here, predefined scans of the water vapour saturation ratios were performed in the diffusion chamber every 30 min. For each scan, $S_{ice}$ steadily increased from 1.2 to 1.8 while the temperature was kept constant. The errors associated to temperature and $S_{ice}$ inside the diffusion chamber were derived from the uncertainty of the thermocouples attached to the instrument walls (±0.5 K)[59].

## Determination of particle formation rate
The particle formation rate, $J_{1.7}$, is determined at 1.7-nm mobility diameter (1.4-nm physical diameter), here using a PSM. At 1.7 nm, a particle is normally considered to be above its critical size and, therefore, thermodynamically stable. $J_{1.7}$ is calculated using the flux of the total concentration of particles growing past a specific diameter (here at 1.7 nm), as well as correction terms accounting for aerosol losses owing to dilution in the chamber, wall losses and coagulation. Details were described previously[47].

## Nucleation model
The nucleation model is on the basis of the thermodynamic model for $H_2SO_4$–$NH_3$ nucleation described in detail previously[18,19]. It is developed from the general dynamic equations[60], to calculate the production and losses for each cluster/particle size to determine the formation rates of the acid–base clusters. For $HNO_3$–$H_2SO_4$–$NH_3$ nucleation, we simplify the model simulations by extrapolating nano-Köhler-type activation by nitric acid and ammonia to clusters down to sulfuric acid trimers. Eighty size bins, ranging from one ammonium sulfate cluster to 300 nm, are used to capture the evolution of the size and composition of polydisperse particles.

In brief, we calculate the equimolar condensation flux of nitric acid and ammonia on the basis of the supersaturation of gas-phase nitric acid and ammonia over particle-phase ammonium nitrate[39,60]:

$$\Phi_i^v = k_c \left[ C_i^v - a_i C_i^0 \right] \tag{2}$$

in which $\Phi_i^v$ is the net condensation flux of nitric acid or ammonia, with vapour concentration $C_i^v$ and saturation concentration $C_i^0$. The term $a_i$ is the activity of species $i$ at the condensed-phase surface of the particle and $k_c$ is the condensation sink for vapours resulting from interaction with particles. The saturation concentrations of nitric acid and ammonia are estimated on the basis of the dissociation constant $K_p$ (ref. [60]). When the vapours are unsaturated, particle-phase ammonium nitrate will evaporate to nitric acid and ammonia to reach the equilibrium.

We also include the Kelvin term ($K_{i,p}$) in the simulation to account for the increased activity ($a_i = a'_i K_{i,p}$) of a small curved cluster/particle:

$$K_{i,p} = 10^{(d_{K10}/d_p)} \tag{3}$$

in which $K_{i,p}$ scales with a 'Kelvin diameter' ($d_{K10}$) for decadal change and $d_p$ is the diameter of the small cluster/particle. The Kelvin diameter for ammonium nitrate is estimated to be 5.3 nm by fitting the data from previous CLOUD experiments according to:

$$S = 10^{(d_{K10}/d_{act})} \tag{4}$$

in which $S$ is the saturation ratio, calculated by means of dividing the product of measured concentrations of nitric acid and ammonia by the dissociation constant $K_p$ and $d_{act}$ is the activation diameter, at which the thermodynamic energy barrier for condensation is overcome and particles start to grow rapidly.

## Determination of ice nucleation ability
During the experiments, aerosol particles were continuously sampled from the CLOUD chamber into the mINKA ice nucleation instrument, using an actively cooled sampling line for a consistent temperature profile. Particles were then subject to well-controlled ice supersaturated conditions; the ones that nucleated ice were selectively detected and counted by an optical particle counter (custom-modified Climet CI-3100, lower detection limit of about 1 μm) located at the outlet of the instrument. Background ice crystals were quantified before each saturation scan (for 2 min) and subtracted from the total ice number concentration of the corresponding measurement. The fraction of INP ($f_{ice}$) was calculated as the ratio of ice crystals number concentration to the total number of particles larger than 10 nm in diameter. The ice nucleation active surface site density ($n_s$)[61] was calculated as the ratio of ice number concentration to the total surface area of particles larger than 10 nm in diameter. The overall uncertainty of $n_s$ is estimated to be ±40% (ref. [24]). Particle number and surface area concentrations were measured by the SMPS described in the 'Instrumentation' section.

In Extended Data Fig. 4, we provide a detailed summary of the measurement data recorded during the 'hotspot condition' experiment shown in Fig. 4a, in which we investigated the heterogeneous crystallization and ice nucleation ability of ammonium nitrate/sulfate particles produced directly from new particle formation. We first formed pure ammonium nitrate particles through nucleation of nitric acid and

ammonia vapours at 223 K and 15–30% relative humidity (over liquid water). When the evolution of the particle size distribution (Extended Data Fig. 4a) levelled off at a median diameter of around 100 nm, we turned on the UV lights and progressively injected $SO_2$ at 03:33 to gradually increase sulfuric acid concentration (Extended Data Fig. 4b). Consequently, in Extended Data Fig. 4c, aerosol mass spectrometer measurements show that particle composition was dominated by ammonium nitrate over the course of the experiment, whereas sulfate appeared approximately 1 h after the injection of $SO_2$. Finally, we show ice nucleation measurements in Extended Data Fig. 4d. Each vertical trajectory represents a saturation ratio scan in mINKA, colour-coded by the measured ice active fraction ($f_{ice}$). In each scan, we use a horizontal black dash to indicate an ice onset threshold corresponding to $f_{ice}$ of $10^{-3}$. Circles indicate the corresponding scans shown in Fig. 4a.

When the particulate sulfate-to-nitrate molar ratio is smaller than 0.0001, the ice nucleation threshold is detected at an ice saturation ratio ($S_{ice}$) of about 1.6, consistent with the homogeneous freezing threshold of aqueous solution droplets[62]. This finding shows that, if particles presented as absolutely pure ammonium nitrate ($NH_4NO_3$), they would exist as supercooled liquid droplets even at very low relative humidity, consistent with previous studies[22,63]. As the particulate sulfate-to-nitrate molar ratio gradually increases to about 0.017, the ice nucleation onset shifts to a lower $S_{ice}$ of 1.2, caused by heterogeneous ice nucleation on crystalline ammonium nitrate particles[23]. Crystalline salts are known to be efficient INP at low temperatures when their deliquescence occurs at higher relative humidity compared with the humidity range of their heterogeneous ice nucleation activity[64]. The fact that the addition of sulfate can promote the crystallization of ammonium nitrate has already been observed in previous studies with particles nebulized in large sizes (around 1 μm) from bulk solutions of ammonium nitrate/sulfate[6,23,65]. But it is evidenced here for the first time in an in situ particle nucleation and crystallization experiment representative of upper tropospheric conditions.

## Particle formation rate parameterization

According to the first nucleation theorem for multicomponent systems[25], we parameterize the particle formation rates ($J_{1.7}$) for the $HNO_3$–$H_2SO_4$–$NH_3$ nucleation scheme with the empirical formula:

$$J_{1.7} = k\,[H_2SO_4]^a\,[HNO_3]^b[NH_3]^c \tag{5}$$

in which vapour concentrations are in units of $cm^{-3}$ and $k$, $a$, $b$ and $c$ are free parameters. This method has been validated by previous observations that the particle formation rates ($J_{1.7}$) vary as a product of power-law functions of nucleating vapours. For example, $J_{1.7}$ for ternary sulfuric acid, ammonia (and water) nucleation follows a cubic dependency on sulfuric acid[8] and a linear[8] or quadratic[19] dependency on ammonia; $J_{1.7}$ for multicomponent nucleation of sulfuric acid, biogenic oxidized organics and ammonia follows a quadratic dependency on sulfuric acid, a linear dependency on both organics[66] and ammonia[11]. The prefactor $k$ accounts for effects of external conditions, such as temperature and relative humidity, thus differs in different environments.

To isolate variables, here we fit the power-law exponents for sulfuric acid, nitric acid and ammonia, respectively, to the dataset of experiments in which only the corresponding vapour concentration was varied. The red triangles, blue circles and yellow squares in Extended Data Fig. 5a–c (same experiments in Extended Data Fig. 1, Fig. 1 and Extended Data Fig. 2), respectively, show that $J_{1.7}$ depends on $[H_2SO_4]^3$ for sulfuric acid between $2.6 \times 10^5$ and $2.9 \times 10^6\ cm^{-3}$ (or 0.008 and 0.09 pptv), on $[HNO_3]^2$ for nitric acid between $2.3 \times 10^8$ and $1.7 \times 10^9\ cm^{-3}$ (or 7 and 52 pptv) and on $[NH_3]^4$ for ammonia between $1.7 \times 10^8$ and $4.9 \times 10^8\ cm^{-3}$ (or 5 and 15 pptv). The third power exponent for sulfuric acid is consistent with previously reported parameterizations for ternary $H_2SO_4$–$NH_3$ nucleation[8,19]. The fourth power exponent for ammonia, however, is larger than those in ternary[8,19] or multicomponent systems[11], which

emphasizes the critical role of ammonia and suggests further bonding between ammonia and nitric acid molecules in the nucleating clusters. Next, we verify the exponents by refitting the product of $[H_2SO_4]^3$, $[HNO_3]^2$ and $[NH_3]^4$ to the full dataset. Extended Data Fig. 5d shows good consistency ($R^2 = 0.9$) of the parameterization among the three experiments, with a slope of $3.4 \times 10^{-71}\ s^{-1}\ cm^{24}$ being the prefactor $k$:

$$J_{1.7} = 3.4 \times 10^{-71}[H_2SO_4]^3[HNO_3]^2[NH_3]^4 \tag{6}$$

This parameterization is representative of new particle formation in the Asian monsoon upper troposphere because our experimental conditions of 223 K and 25% relative humidity, as well as concentrations of sulfuric acid[67,68] and nitric acid[69,70], are within the upper tropospheric range, with ammonia[5,6] typical of Asian monsoon regions. One caveat, however, is that the cosmic radiation was at the ground level in our chamber, as shown with grey dot-dashed horizontal line in Extended Data Fig. 5d. The ion-pair production rate can be up to ten times higher in the ambient upper troposphere[71], potentially leading to further enhancement of $J_{1.7}$ by ion-induced nucleation, although the neutral channel dominates in our experiments.

## Estimated temperature dependence of the particle formation rate

We did not cover the full temperature range in the upper troposphere, instead focusing on 223 K. However, to make the parameterization in the previous section more applicable for model simulations while not overstating the role of this mechanism, we provide some constraints on the temperature dependence of $J_{1.7}$ for $HNO_3$–$H_2SO_4$–$NH_3$ nucleation. Broadly, it is certain that particle formation involving $HNO_3$ will have a strong temperature dependence, becoming much slower as $T$ increases.

We first present the temperature dependence of $J_{1.7}$ for pure $HNO_3$–$NH_3$ nucleation with the expression:

$$J_{1.7} = k(T)f([HNO_3], [NH_3]) \tag{7}$$

in which $k(T)$ is an empirical temperature-dependent rate constant and has the Arrhenius form

$$k(T) = e^{\left(-\frac{1}{T}\frac{E}{R}\right)}, \tag{8}$$

in which $T$ is the absolute temperature (in Kelvin), $E$ is the activation energy and $R$ is the universal gas constant. $f([HNO_3],[NH_3])$ is a function of the ammonia and nitric acid concentrations (including the pre-exponential factor and free-fitting parameters). This expression is then fitted to the dataset in Fig. 3c in our previous study[16], in which $J_{1.7}$ were measured with only nitric acid, ammonia and water vapours added to the chamber, and the temperature was progressively decreased from 258 K to 249 K. Because the ammonia and nitric acid concentrations were kept almost constant during the temperature transition, we treat the $f([HNO_3],[NH_3])$ term as a constant to reduce the degrees of freedom. This expression with its two free parameters leads to a good agreement with the data, $R_2 = 0.96$. And the fitted $-E/R$ and $f([HNO_3],[NH_3])$ are 14,000 K and $3.2 \times 10^{-26}$, respectively.

Next, we apply the same $k(T)$ term to the $HNO_3$–$H_2SO_4$–$NH_3$ parameterization (equation (9)), assuming that the multicomponent nucleation follows a similar temperature dependence:

$$J_{1.7} = 2.9 \times 10^{-98} e^{\left(\frac{14,000}{T}\right)}[H_2SO_4]^3[HNO_3]^2[NH_3]^4 \tag{9}$$

Although this temperature-dependent parameterization may not be the final description of this process, it tracks the trend of $J_{1.7}$ well. In the event of $4 \times 10^6\ cm^{-3}\ H_2SO_4$, $1.5 \times 10^9\ cm^{-3}\ HNO_3$ and $5 \times 10^8\ cm^{-3}$ $NH_3$, the multicomponent nucleation is quenched ($J_{1.7} < 0.01\ cm^{-3}\ s^{-1}$) above 268 K. This is consistent with the observations that nitric acid

and ammonia only contribute to the growth of ammonium sulfate particles at 278 K (ref. [16]). At 223 K, the parameterized $J_{1.7}$ is 306 cm$^{-3}$ s$^{-1}$, matching our measurement in Fig. 2. And for the temperature in the upper troposphere and lower stratosphere (198 K), the parameterized $J_{1.7}$ is $8 \times 10^5$ cm$^{-3}$ s$^{-1}$, which is still much slower than its kinetic limit of about $10^9$–$10^{10}$ cm$^{-3}$ s$^{-1}$.

### The EMAC global model

The ECHAM/MESSy Atmospheric Chemistry (EMAC) model is a numerical chemistry and climate simulation system that includes sub-models describing tropospheric and middle atmosphere processes and their interaction with oceans, land and human influences[72]. It uses the second version of the Modular Earth Submodel System (MESSy2) to link multi-institutional computer codes. Atmospheric circulation is calculated by the 5th generation of the European Centre Hamburg general circulation model (ECHAM5 (ref. [73])) and atmospheric chemical kinetics are solved for every model time step. For the present study, we applied EMAC (ECHAM5 version 5.3.02, MESSy version 2.54.0) in the T42L31ECMWF-resolution, for example, with a spherical truncation of T42 (corresponding to a quadratic Gaussian grid of approximately 2.8° by 2.8° in latitude and longitude) with 31 vertical hybrid pressure levels up to 10 hPa. EMAC uses a modal representation of aerosols dynamics (GMXe) that describes the aerosol size distribution as seven interacting log-normal distributions, of which four modes are soluble and three modes are insoluble. New particles are added directly to the nucleation mode. The applied model setup comprises the sub-model New Aerosol Nucleation (NAN) that includes new parameterizations of aerosol particle formation rates published in recent years[74]. These parameterizations include ion-induced nucleation. The ion-pair production rate, needed to calculate the ion-induced or ion-mediated nucleation, is described using the sub-model IONS, which provides ion-pair production rates[74].

### The TOMCAT global model

The TOMCAT model is a global 3D offline chemical transport model[75,76]. It is run at approximately 2.8° spatial resolution, such as EMAC on a T42 grid, driven by ECMWF ERA-Interim reanalysis meteorological fields for the year 2008. We also used 31 hybrid sigma-pressure levels from the surface to 10 hPa. The dissolved fraction of gases in cloud water is calculated by means of an equilibrium Henry's law approach and set to zero for temperatures below −20 °C. The model includes GLOMAP aerosol microphysics[77] with nitrate and ammonium from the HyDIS solver[78] and the representation of new particle formation used by Gordon et al.[3]. The HyDIS solver adopts a sophisticated approach to the dissolution of nitric acid and ammonia into the aerosol phase that is a hybrid between a dynamic representation of the process, which accounts for the time needed for mass transport, and an equilibrium representation, which does not[78]. The main limitation of the solver is that it assumes all aerosol particles are liquid, which is probably a poor approximation in cold, dry conditions frequently found in the upper troposphere.

### The cloud trajectories framework

We conducted a sensitivity study on ammonia transport processes and estimated the fraction remaining of ammonia vapour after convection from the boundary layer to the upper troposphere, using a cloud trajectories framework described in detail in Bardakov et al.[79,80]. In brief, trajectories from a convective system simulated with the large-eddy simulation (LES) model MIMICA[81] were extracted and a parcel representing the cloud outflow was selected for further analysis (Extended Data Fig. 8a). The meteorological profiles and clouds microphysics scheme used here were the same as in Bardakov et al.[80], producing altitude-dependent distributions of water and ice hydrometeors depicted in Extended Data Fig. 8. Partitioning of gas between vapour and aqueous phase along the trajectory was calculated on the basis of Henry's law constant adjusted to a cloud pH, $H^* = H \times 1.7 \times 10^{(9-pH)}$ following the expression for ammonia from Seinfeld and Pandis[60].

We then investigated the factors governing ammonia transport through the simulated convective system by varying: (1) the pH for the liquid water hydrometeors (Extended Data Fig. 8c); (2) the total amount of water in the system (Extended Data Fig. 8d); (3) the retention of ammonia molecules by the ice hydrometeors (Extended Data Fig. 8e). In our base-case simulation, the pH was assumed to have an altitude-dependent profile, reflecting the higher abundance of acids close to the surface and ranging from 4.5 to 5, in accordance with the representative pH values in the EMAC simulation. The base-case water content was as in Bardakov et al.[80] and the ice retention coefficient 0.05 in accordance with Ge et al.[13], with no further uptake on ice.

### Atmospheric interpretation

This work focuses on the Asian monsoon region in part because this region is fairly extensive, but also because ammonia concentrations measured in this region are by far the highest in the upper troposphere. Although we frame this synergistic $HNO_3$–$H_2SO_4$–$NH_3$ nucleation in a scenario that suits the Asian monsoon upper troposphere, the physics applies more broadly − the colder the conditions are, the more important this mechanism is likely to be. To explore the importance of this synergistic nucleation to the atmosphere, we combine our experimental results, cloud resolving modelling and global-scale chemical transport modelling. On the basis of these constraints, the rate-limiting elements of new particle formation seem to be convective transport of ammonia and the production rate of particles in the mixing zone between convective outflow and the background upper free troposphere; however, confirmation of this will require extensive field and modelling studies.

Generally, nitric acid ranges between about $10^8$ and $10^9$ cm$^{-3}$ (refs. [14,15]) and sulfuric acid between about $10^5$ and $10^6$ cm$^{-3}$ (refs. [82,83]) in the tropical upper troposphere. The typical acid-excess conditions leave the principal uncertainty being ammonia levels, which are not yet well constrained. Although satellite-based ammonia measurements have provided a spatial distribution on a global scale, they are limited to cloud-free areas owing to blockage of the ammonia signal by optically thick clouds. However, deep convection followed by cloud glaciation may be a major source of upper tropospheric ammonia. This process may then not be captured by satellites as it occurs near clouds, with short time duration and high spatial heterogeneity. This may also explain why the in situ-measured ammonia concentrations are up to 40 times higher than those from satellite measurements[6].

Ammonia has no known chemical source in the atmosphere but is instead transported by cloud processes from the surface, whereas nitric acid and sulfuric acid vapours are formed primarily by out-of-cloud oxidation. Consequently, it is probable that this synergistic nucleation occurs initially in the outflow of convective clouds, in which the released ammonia mixes with pre-existing (background) nitric acid and sulfuric acid. Subsequently, as ammonia is titrated over several $e$-folding times (governed by the condensation sink in this mixing zone) and the outflow air fully mixes with the background air, nucleation conditions will shift from the ammonia-rich regime to the ammonia-limited regime. These highly dynamic processes are thus the key to constraining the climatic effects of this synergistic nucleation in Asian monsoon and potentially other convective regions. Nevertheless, current ambient measurements confirm the presence of ample ammonia, as well as particles comprised largely of ammonium nitrate[4], and our experiments show that synergistic $HNO_3$–$H_2SO_4$–$NH_3$ nucleation is a viable mechanism for new particle formation in the Asian monsoon upper troposphere. As global ammonia emissions continue to increase owing to agricultural growth and the warmer climate[84,85], the importance of this particle formation mechanism will increase.

Further, as there is almost no in situ composition measurement of clusters or newly formed particles in the upper troposphere, we can

only infer the major particle formation pathway from indirect evidence such as composition of precursor vapours or larger particles. Previously established mechanisms include binary and ternary sulfuric acid nucleation, which drive new particle formation over marine or anthropogenically influenced regions[1,4,86,87], nucleation by oxygenated organics, which dominates over pristine vegetated areas such as the Amazon basin[2,10,88], and nucleation by iodine oxidation products, which may be especially important in marine convection[89,90]. Over the Asian monsoon regions, however, mixed emissions of both inorganic and organic vapours may well complicate the particle formation mechanism. However, it has been demonstrated that ammonium nitrate can often explain more than half of the particulate volume in the upper troposphere[6]. This means that the $HNO_3$–$NH_3$ concentration is probably higher than the sum of all other condensable vapours (that is, sulfuric acid and oxygenated organics). And given that $HNO_3$–$H_2SO_4$–$NH_3$ nucleation is orders of magnitude faster than binary and ternary sulfuric acid nucleation at observed ammonia levels, we therefore infer that synergistic $HNO_3$–$H_2SO_4$–$NH_3$ nucleation is a major particle formation pathway in the Asian monsoon upper troposphere. It seems unlikely that this inorganic pathway and the organic pathways are antagonistic in growth, and without strong indications otherwise, it seems probable that they are more or less additive for nucleation itself. However, to further investigate interactions between different nucleation schemes, we would rely on further information on the source and identity of organic vapours that are present in the Asian monsoon upper troposphere.

## Data availability

The full dataset shown in the figures is publicly available at https://doi.org/10.5281/zenodo.5949440. Source data are provided with this paper.

## Code availability

The EMAC (ECHAM/MESSy) model is continuously further developed and applied by a consortium of institutions. The use of MESSy and access to the source code is licensed to all affiliates of institutions that are members of the MESSy Consortium. Institutions can become a member of the MESSy Consortium by signing the MESSy Memorandum of Understanding. More information can be found on the MESSy Consortium website (https://www.messy-interface.org). All code modifications presented in this paper will be included in the next version of MESSy. The TOMCAT model (http://homepages.see.leeds.ac.uk/~lecmc/tomcat.html) is a UK community model. It is available to UK (or NERC-funded) researchers who normally access the model on common facilities or who are helped to install it on their local machines. As it is a complex research tool, new users will need help to use the model optimally. We do not have the resources to release and support the model in an open way. Any potential user interested in the model should contact Martyn Chipperfield. The model updates described in this paper are included in the standard model library. The cloud trajectories model is publicly available at https://doi.org/10.5281/zenodo.5949440. Codes for conducting the analysis presented in this paper can be obtained by contacting the corresponding author, Neil M. Donahue (nmd@andrew.cmu.edu).

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

**Acknowledgements** We thank the European Organization for Nuclear Research (CERN) for supporting CLOUD with important technical and financial resources. This research has received funding from the US National Science Foundation (nos. AGS-1801574, AGS-1801897, AGS-1602086, AGS-1801329, AGS-2132089 and AGS-1801280), the European Union's Horizon 2020 programme (Marie Skłodowska-Curie ITN no. 764991 'CLOUD-MOTION'), the European Commission, H2020 Research Infrastructures (FORCeS, no. 821205), the European Union's Horizon 2020 research and innovation programme (Marie Skłodowska-Curie no. 895875 'NPF-PANDA'), a European Research Council (ERC) project ATM-GTP contract (no. 742206), an ERC-CoG grant INTEGRATE (no. 867599), the Swiss National Science Foundation (nos. 200021_169090, 200020_172602 and 20FI20_172622), the Academy of Finland ACCC Flagship (no. 337549), the Academy of Finland Academy professorship (no. 302958), the Academy of Finland (nos. 1325656, 316114 and 325647), Russian MegaGrant project 'Megapolis – heat and pollution island: interdisciplinary hydroclimatic, geochemical and ecological analysis' (application reference 2020-220-08-5835), Jane and Aatos Erkko Foundation 'Quantifying carbon sink, CarbonSink+ and their interaction with air quality' INAR project, Samsung PM2.5 SRP, Prince Albert Foundation 'the Arena for the gap analysis of the existing Arctic Science Co-Operations (AASCO)' (no. 2859), the German Federal Ministry of Education and Research (CLOUD-16 project nos. 01LK1601A and 01LK1601C), the Knut and Alice Wallenberg Foundation Wallenberg Academy Fellows project AtmoRemove (no. 2015.0162), the Portuguese Foundation for Science and Technology (no. CERN/FIS-COM/0014/2017) and the Technology Transfer Project N059 of the Karlsruhe Institute of Technology (KIT). The FIGAERO-CIMS was supported by a Major Research Instrumentation (MRI) grant for the US NSF AGS-1531284, as well as the Wallace Research Foundation. The computations by R.Bardakov were performed on resources provided by the Swedish National Infrastructure for Computing (SNIC) at the National Supercomputer Center (NSC). I.R. thanks the Max Planck Society for a sabbatical award. M.W. thanks Siebel Scholars Foundation for financial support.

**Author contributions** M.W., B.B., J.K. and N.M.D. planned the experiments. M.W., B.B., G.M., B.R., B.S., X.-C.H., J.S., W.S., R.M., B.L., H.L., H.E.M., F.A., P.B., Z.B., L.C., L.-P.D.M., J.D., H.F., L.G.C., M.G., R.G., V.H., A.K., K.L., V.M., D.M., S.M., R.L.M., B.M., T.M., A.O., T.P., M.P., A.A.P., A.P., M.Simon, Y.S., A.T., N.S.U., F.V., R.W., D.S.W., S.K.W., A.W., Y.W., M.Z.-W., M.Sipilä, P.M.W., A.H., U.B., M.K., R.C.F., J.C., R.V., I.E.-H., J.K., K.K. O.M., S.S. and N.M.D. prepared the CLOUD facility or measuring instruments. M.W., B.B., G.M., B.R., B.S., X.-C.H., J.S., W.S., R.M., B.L., H.E.M., A.A., L.C., L.G.C., M.G., M.H., V.H., J.E.K., N.G.A.M., D.M., R.L.M., B.M., A.R., M.Schervish, M.Simon, A.T., N.S.U., F.V., D.S.W., S.K.W., A.W., M.Z.-W., P.M.W., J.K. and K.K. collected the data. M.W., M.X., B.B., G.M., B.R., B.S., R.Bardakov, J.S., W.S., L.D., R.Baalbaki, B.L., D.S.W., S.K.W., A.W., I.R., T.C. and N.M.D. analysed the data. M.W., M.X., B.B., R.Bardakov, X.-C.H., J.S., W.S., R.M., L.D., R. Baalbaki., B.L., H.L., H.E.M., A.M.L.E., H.F., M.H., K.H., A.K., N.S.U., R.W., A.W., A.H., U.B., M.K., R.C.F., J.C., R.V., I.R., H.G., J.L., I.E.-H., D.R.W., T.C., J.K., O.M., S.S. and N.M.D. contributed to the scientific discussion. M.W., B.B., R.Bardakov, W.S., R.M., B.L., K.H., A.K., U.B., R.C.F., J.C., R.V., I.R., H.G., J.L., I.E.-H., T.C., J.K., O.M. and N.M.D. wrote the manuscript.

**Competing interests** The authors declare no competing interests.

**Additional information**
**Correspondence and requests for materials** should be addressed to Neil M. Donahue.

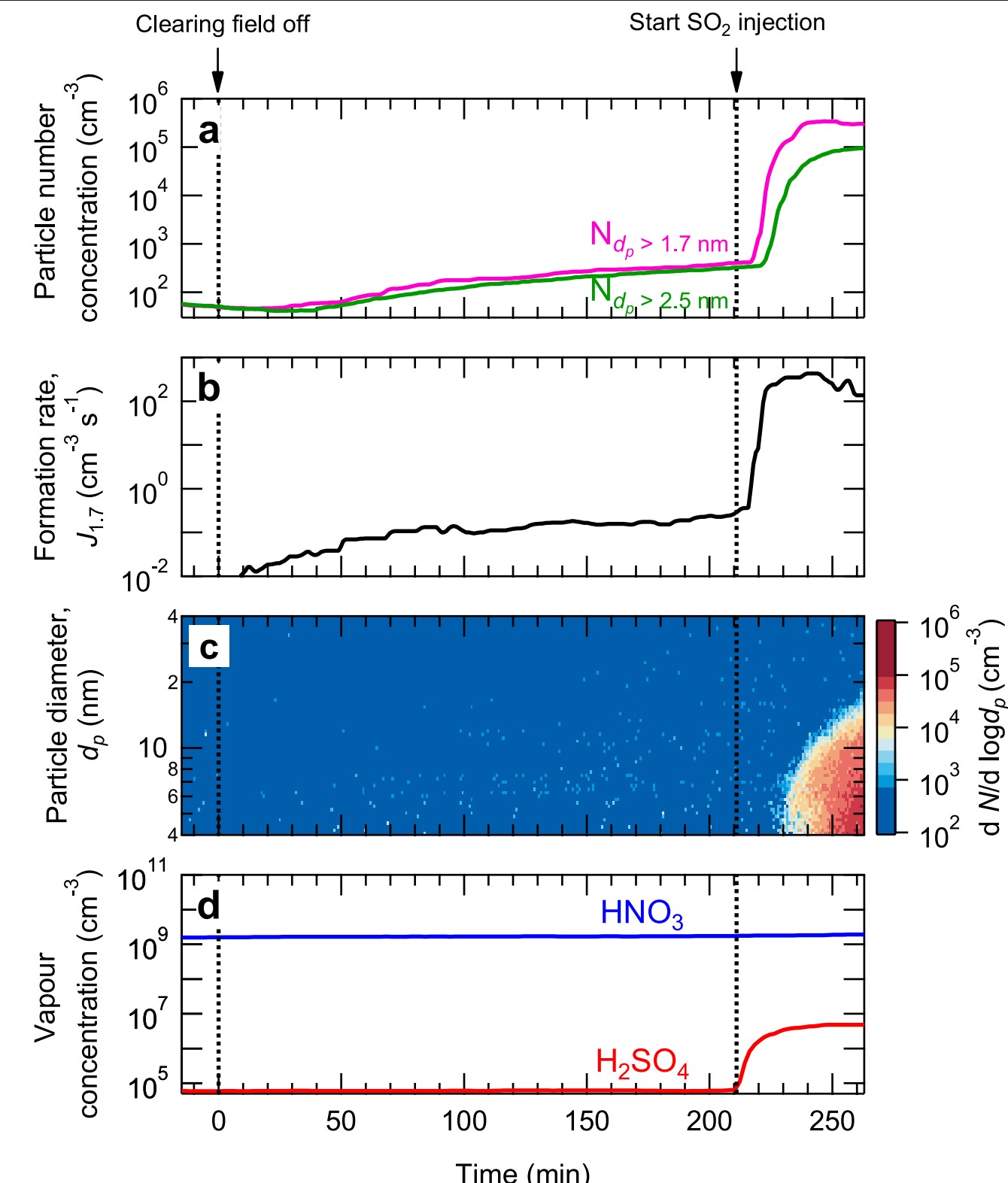

**Extended Data Fig. 1** | See next page for caption.

**Extended Data Fig. 1 | Enhancement of $HNO_3$–$NH_3$ particle formation by sulfuric acid. a**, Particle number concentrations versus time at mobility diameters >1.7 nm (magenta) and >2.5 nm (green). The solid magenta trace is measured by a $PSM_{1.7}$ and the solid green trace is measured by a $CPC_{2.5}$. The fixed experimental conditions are about $6.5 \times 10^8$ cm$^{-3}$ $NH_3$, 223 K and 25% relative humidity. **b**, Particle formation rate versus time at 1.7 nm ($J_{1.7}$), measured by a PSM. **c**, Particle size distribution versus time, measured by an SMPS. **d**, Gas-phase nitric acid and sulfuric acid versus time, measured by an I$^-$ CIMS and a $NO_3^-$ CIMS, respectively. We started the experiment by oxidizing $NO_2$ to produce $1.6 \times 10^9$ cm$^{-3}$ $HNO_3$ in the presence of about $6.5 \times 10^8$ cm$^{-3}$ ammonia. At time = 0 min, we turned off the high-voltage clearing field to allow the ion concentration to build up to a steady state between GCR production and wall deposition. The presence of ions (GCR condition) induces slow $HNO_3$–$NH_3$ nucleation, followed by relatively fast particle growth by nitric acid and ammonia condensation. We thus observe formation of both 1.7-nm and 2.5-nm particles by about one order of magnitude in about 3.5 h, with a slower approach to steady state because of the longer wall deposition time constant for the larger particles. Then, we increased $H_2SO_4$ in the chamber from 0 to $4.9 \times 10^6$ cm$^{-3}$ by oxidizing progressively more injected $SO_2$ after 211 min, with a fixed production rate of nitric acid and injection rate of ammonia. Subsequently, particle concentrations increase by three orders of magnitude within 30 min. The overall systematic scale uncertainties of ±30% on particle formation rate, −33%/+50% on sulfuric acid concentration and ±25% on nitric acid concentration are not shown.

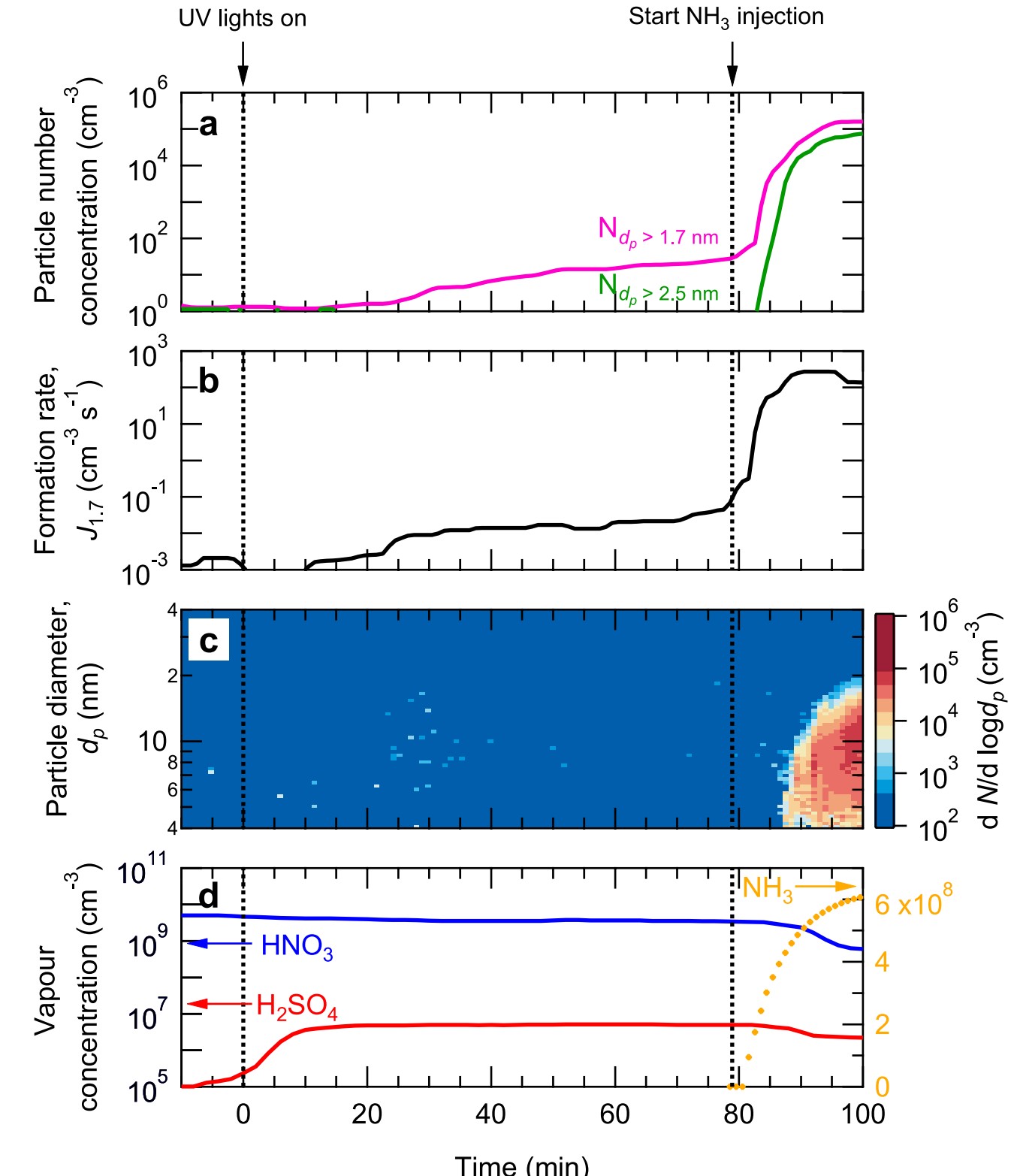

**Extended Data Fig. 2** | See next page for caption.

**Extended Data Fig. 2 | Enhancement of $H_2SO_4$–$HNO_3$ nucleation by ammonia. a**, Particle number concentrations versus time at mobility diameters >1.7 nm (magenta) and >2.5 nm (green). The solid magenta trace is measured by a $PSM_{1.7}$ and the solid green trace is measured by a $CPC_{2.5}$. The fixed experimental conditions are 223 K and 25% relative humidity. **b**, Particle formation rate versus time at 1.7 nm ($J_{1.7}$), measured by a PSM. **c**, Particle size distribution versus time, measured by an SMPS. **d**, Gas-phase nitric acid and sulfuric acid versus time, measured by an I⁻ CIMS and a $NO_3^-$ CIMS, respectively; gas-phase ammonia versus time, calculated with a first-order wall-loss rate. Before the experiment, we cleaned the chamber by rinsing the walls with ultra-pure water, followed by heating to 373 K and flushing at a high rate with humidified synthetic air for 48 h. We started with an almost perfectly clean chamber and only $HNO_3$, $SO_2$ and $O_3$ vapours present at constant levels. Sulfuric acid starts to appear by means of $SO_2$ oxidation soon after switching on the UV lights at time = 0 min, building up to a steady state of $5.0 \times 10^6$ cm⁻³ with the wall-loss timescale of about 10 min. Subsequently, we observe slow formation of 1.7-nm particles, yet they do not reach 2.5 nm during the course of a 2-h period with small growth rates and low survival probability. Then, owing to the injection of ammonia from 0 to around $6.5 \times 10^8$ cm⁻³ into the chamber after 80 min, a sharp increase in the rate of particle formation is observed with a fixed production rate of sulfuric acid and injection rate of nitric acid. The sulfuric acid concentration decreases slightly afterwards, owing to accumulated condensation sink from fast particle growth. The overall systematic scale uncertainties of ±30% on particle formation rate, −33%/50% on sulfuric acid concentration and ±25% on nitric acid concentration are not shown.

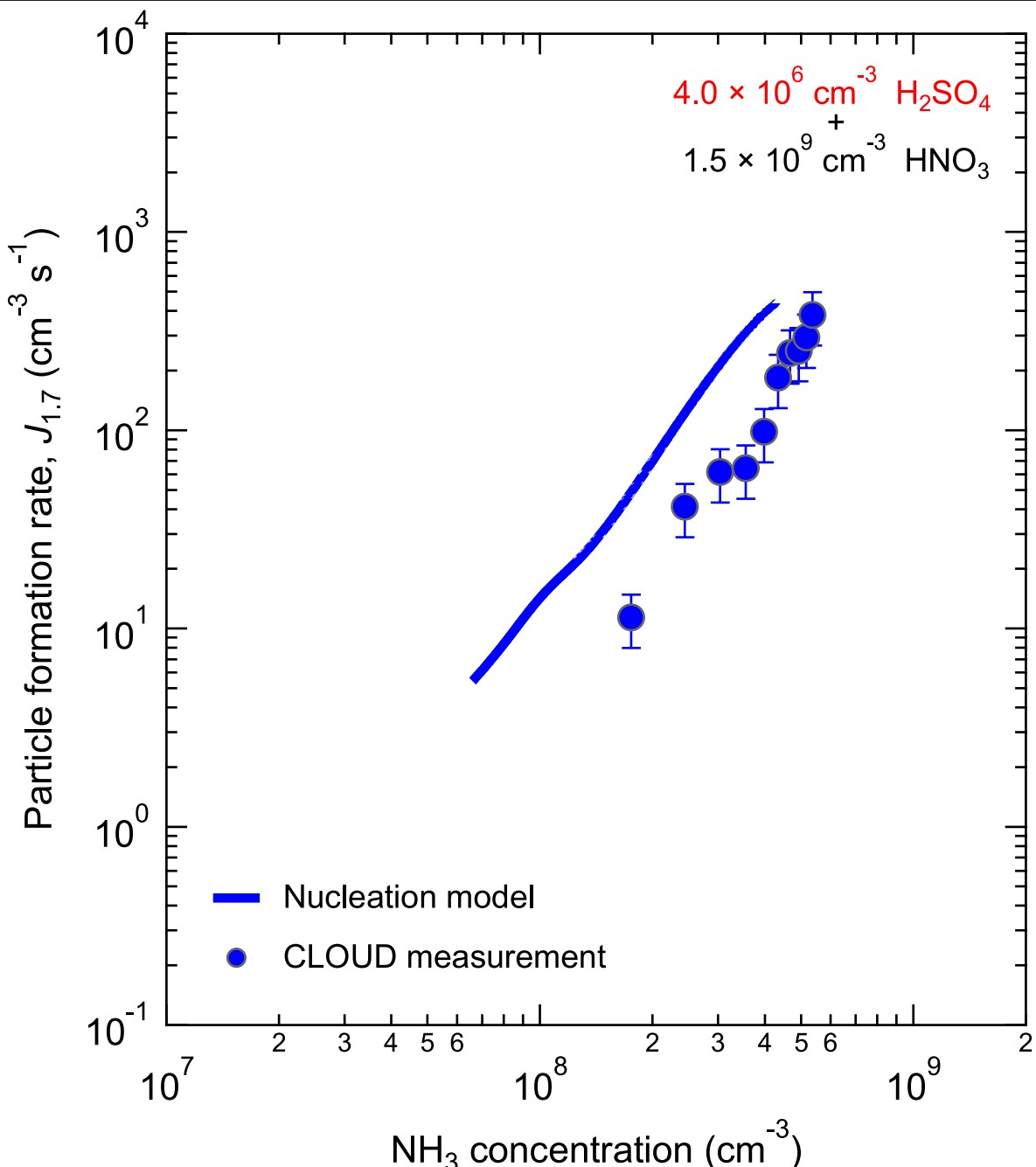

**Extended Data Fig. 3 | Particle formation rates at 1.7 nm ($J_{1.7}$) versus ammonia concentration at 223 K and 25% relative humidity.** Circles are the CLOUD measurements (the same as those in Fig. 2). The curve represents the model simulations on the basis of known thermodynamics and microphysics, including Kelvin effects, for nucleating clusters.

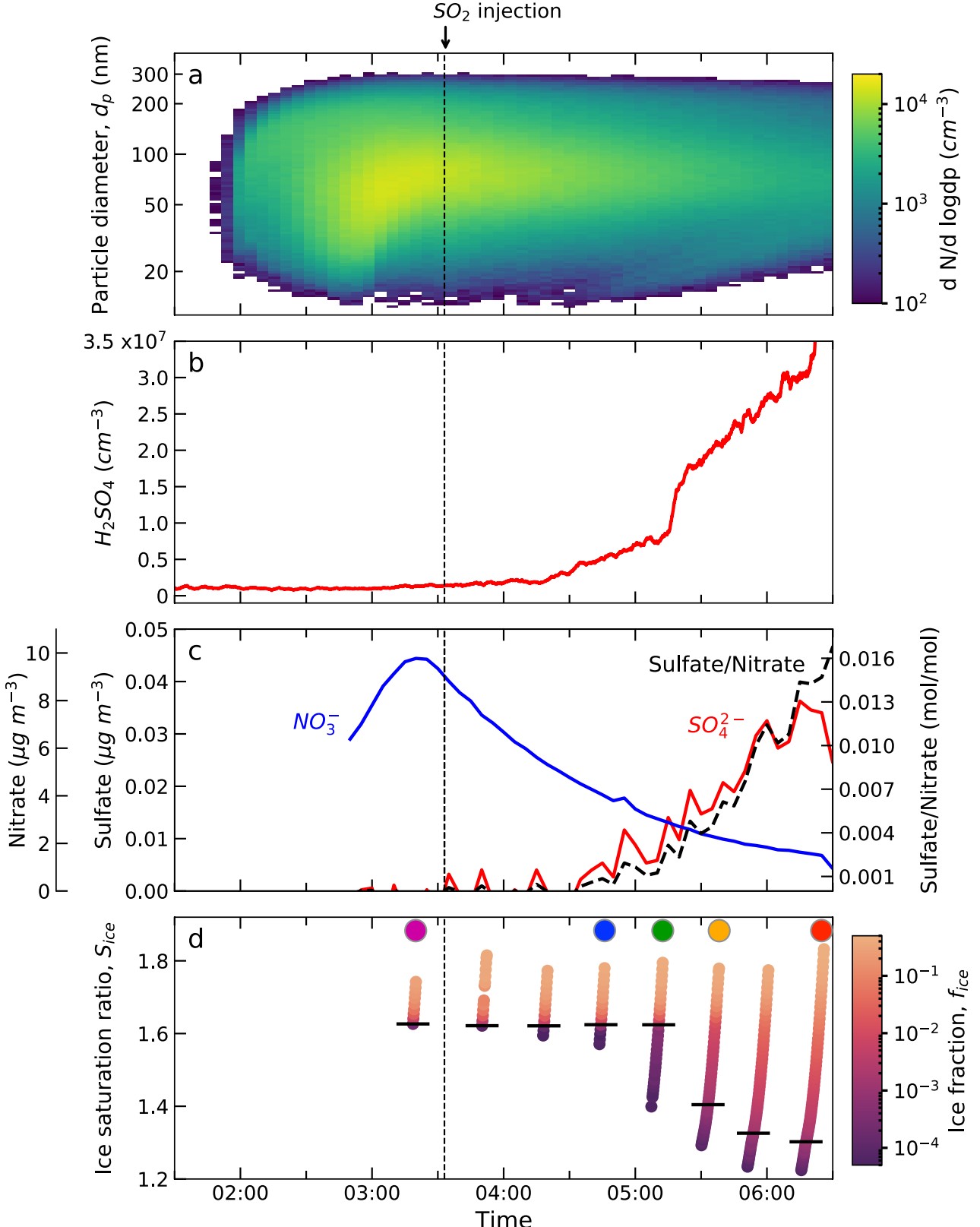

**Extended Data Fig. 4 | Measurement of the ice nucleation ability of**
**HNO₃–H₂SO₄–NH₃ particles versus sulfate-to-nitrate ratio. a**, Particle size
distribution versus time during the experiment, measured by an SMPS.
**b**, Gas-phase sulfuric acid versus time, measured by a nitrate CIMS.

**c**, Particle-phase chemical composition versus time, measured by an AMS.
**d**, Fraction of INP at the nominal temperature of 215 K. The horizontal black
dashes indicate the ice fraction threshold, $f_{ice} = 10^{-3}$. The coloured circles
correspond to the sulfate-to-nitrate ratios shown in Fig. 4a.

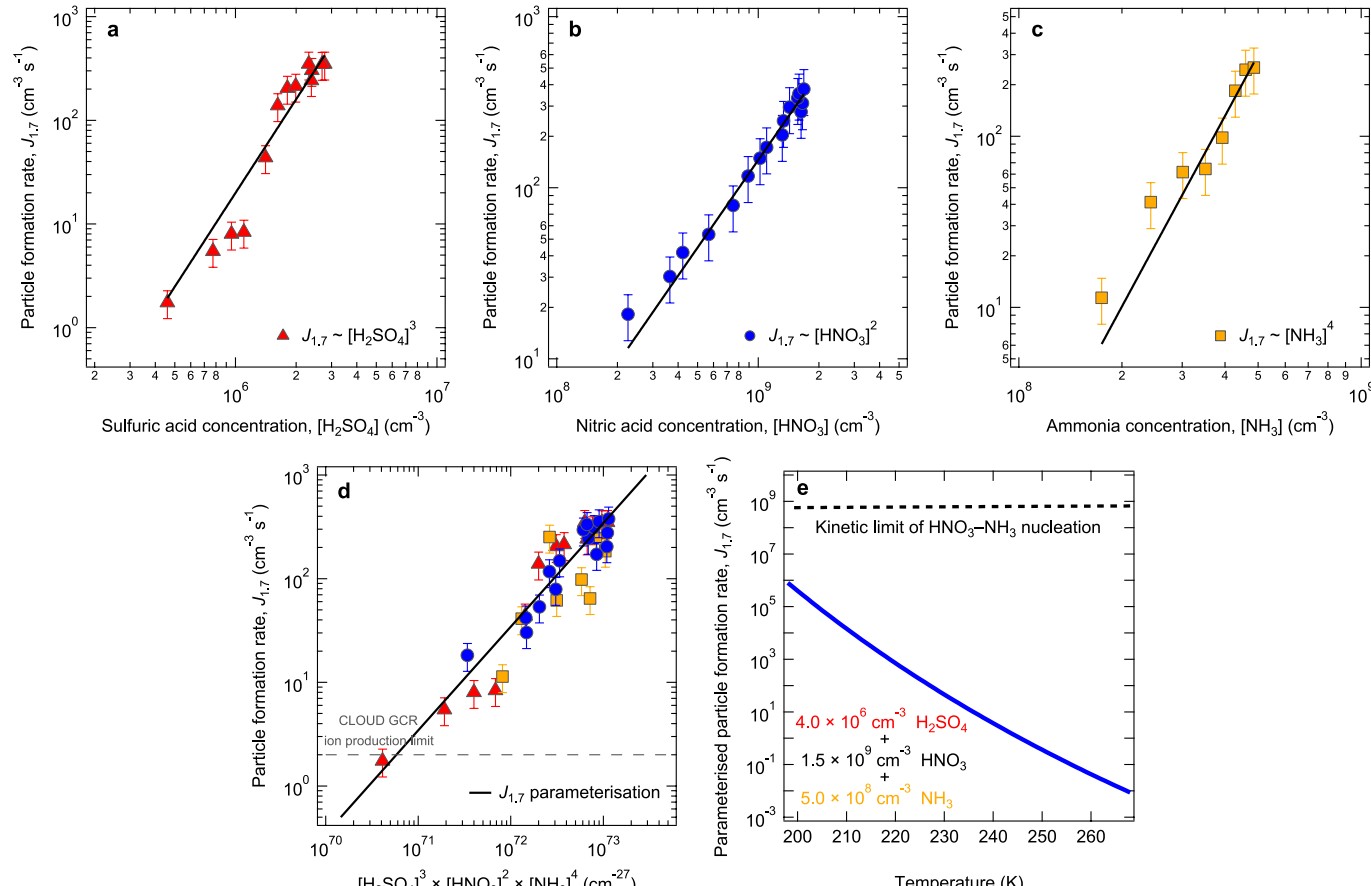

**Extended Data Fig. 5 | Parameterization of the HNO₃–H₂SO₄–NH₃ particle formation rate. a–c** Particle formation rate ($J_{1.7}$) as a function of $H_2SO_4$, $HNO_3$ and $NH_3$ vapour concentrations, respectively, at 223 K and 25% relative humidity. The red triangles, blue circles and yellow squares represent experiments while varying only the concentration of $H_2SO_4$ (Extended Data Fig. 1), $HNO_3$ (Fig. 1) and $NH_3$ (Extended Data Fig. 2), respectively. The $H_2SO_4$ concentration was varied between $4.6 \times 10^5$ and $2.9 \times 10^6$ cm⁻³, $HNO_3$ between $2.3 \times 10^8$ and $1.7 \times 10^9$ cm⁻³ and $NH_3$ between $1.8 \times 10^8$ and $5.1 \times 10^8$ cm⁻³. **d,** The multi-acid–ammonia parameterization (black line) on the basis of equation (6)

with $k = 3.4 \times 10^{-71}$ s⁻¹ cm²⁴. The grey dashed horizontal line shows a maximum of about 2 cm⁻³ s⁻¹ ion-induced nucleation in the CLOUD chamber under GCR conditions, limited by the ion-pair production rate from GCR plus beam-background muons. The bars indicate 30% estimated total error on the particle formation rates, although the overall systematic scale uncertainties of −33%/+50% on sulfuric acid concentration and ±25% on nitric acid concentration are not shown. **e,** Temperature dependence of $J_{1.7}$ for $HNO_3$–$H_2SO_4$–$NH_3$ nucleation (blue curve) on the basis of equation (9) with $k = 2.9 \times 10^{-98}$ e^{14,000/T} s⁻¹ cm²⁴.

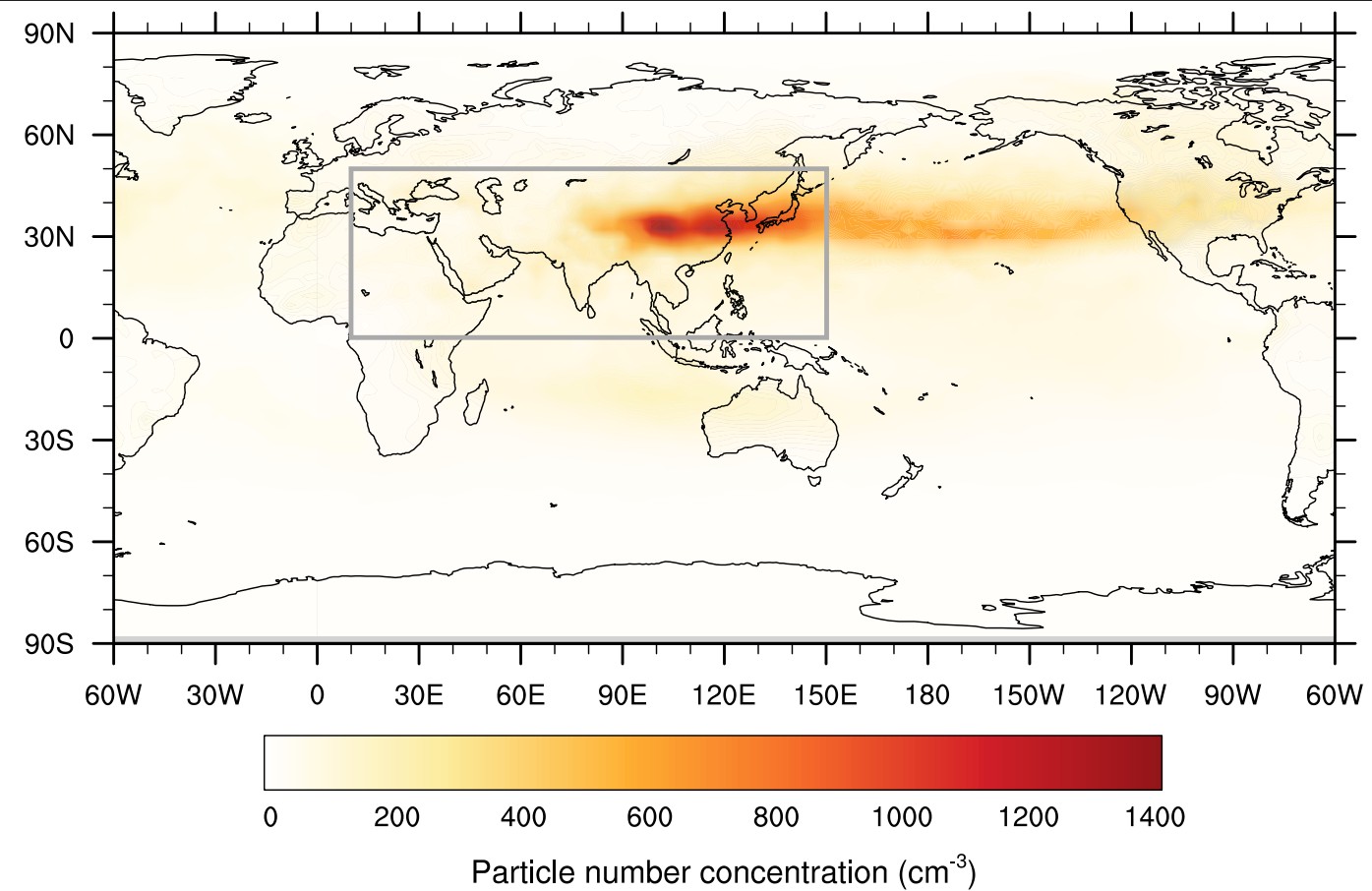

**Extended Data Fig. 6 | Modelled contribution of HNO₃–H₂SO₄–NH₃ nucleation to upper tropospheric particles.** Number concentrations of multi-acid new particles (nucleation mode) at 250-hPa altitude simulated in a global model (EMAC) with efficient vertical transport of ammonia. The particle formation rate is on the basis of the blue dashed curve in Fig. 2 and parameterization shown in Extended Data Fig. 5. The extra particle number concentrations are shown, that is, relative to the same model without multi-acid nucleation. High annually averaged particle numbers are expected in the monsoon region (grey rectangle) and adjacent regions.

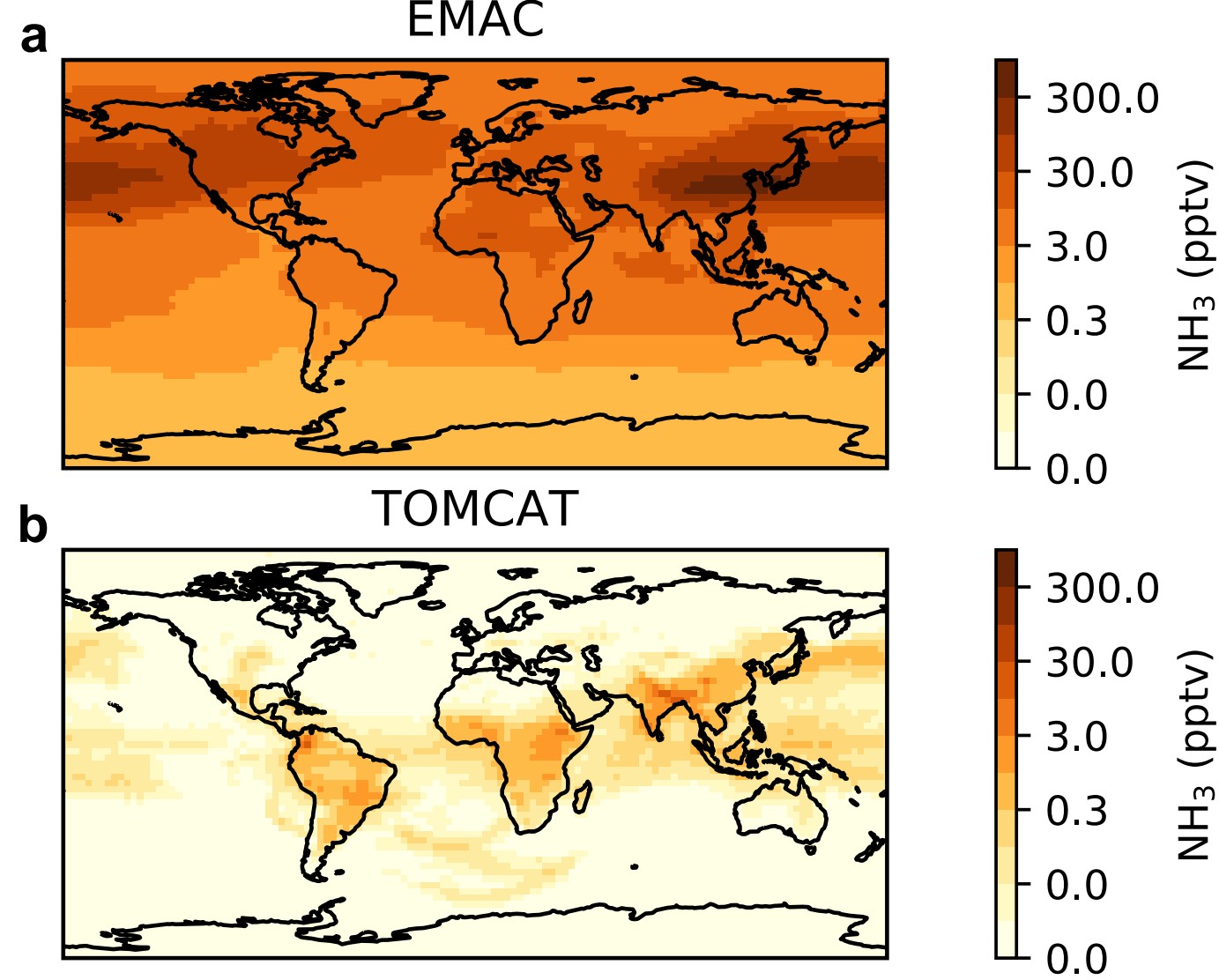

**Extended Data Fig. 7 | Modelled annual mean ammonia mixing ratios at 250 hPa (11 km, about 223 K). a**, The EMAC global model simulations are higher than the MIPAS satellite observations, although consistent with aircraft measurements[5,6]. **b**, The TOMCAT global model predicts much less ammonia (<1 pptv) in the upper troposphere.

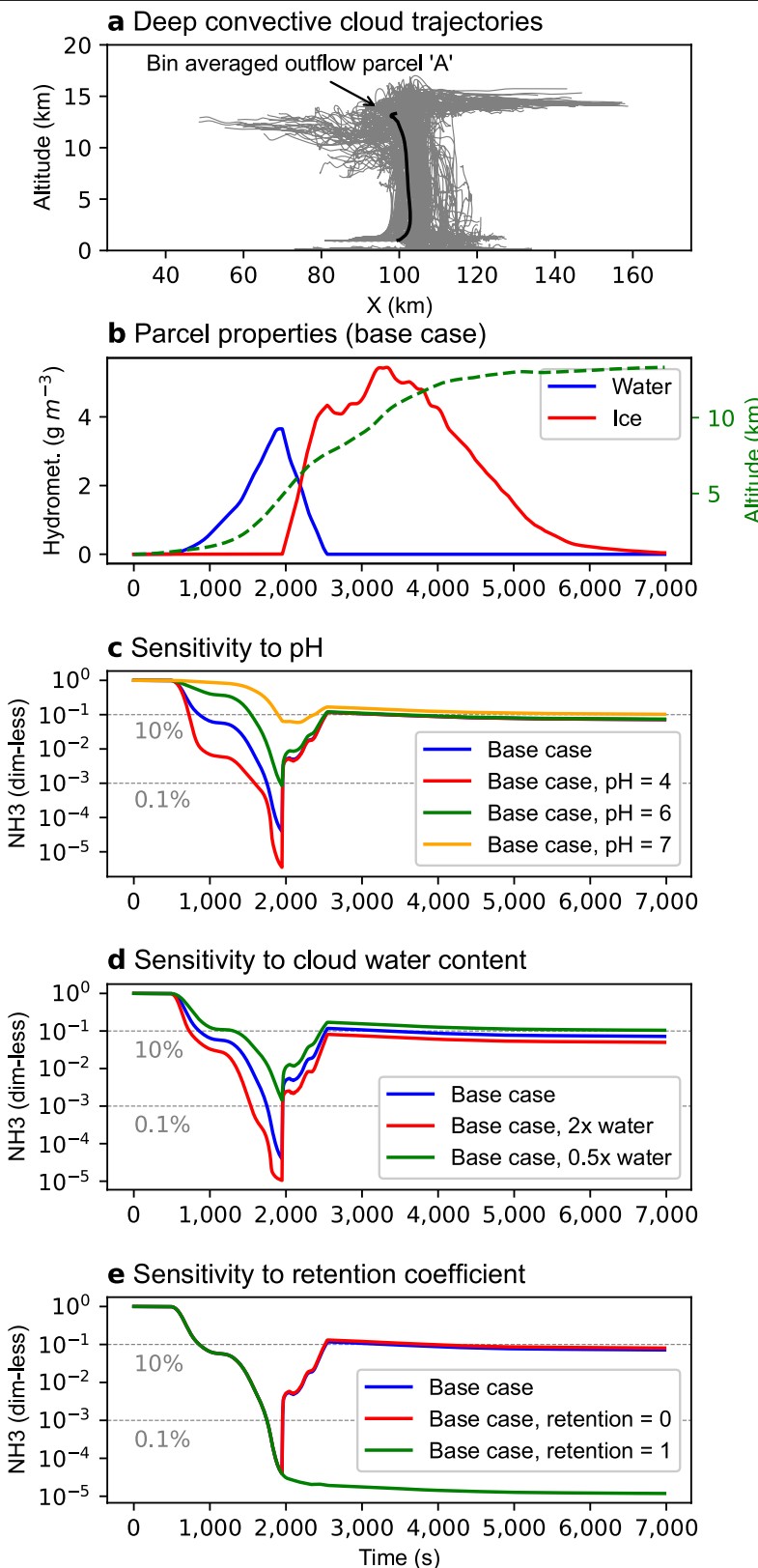

**Extended Data Fig. 8 | Modelled transport of ammonia to the upper troposphere in deep convective clouds. a**, Trajectories of the simulated convective cloud event (grey) and a selected parcel representing a buoyant parcel reaching the upper troposphere (black). **b**, The simulated evolution of parcel A altitude (green dashed trace) and the total mass concentration and phase of the cloud hydrometeors (red and blue curves). **c–e** Sensitivity of the predicted ammonia concentrations within parcel A to cloud water pH, total water amount and retention coefficient (by ice particles) as compared with the base-case simulation (blue trace in all figures).