## [Peer Review File · Nature]

Manuscript Title: Synergistic HNO₃–H₂SO₄–NH₃ upper tropospheric particle formation

Reviewer Comments & Author Rebuttals

Reviewer Reports on the Initial Version:

Referees' comments:

Referee #1 (Remarks to the Author):

Review of “Synergistic particle formation in the upper troposphere by nitric acid, sulfuric acid and ammonia” by M. Wang et al.

The CLOUD chamber experiments, described by the authors, support nitric acid enhancement of sulfuric acid-ammonia in new particle formation. This is an interesting and important result that sheds new light on the multicomponent nucleation mechanisms controlling new particle formation and their subsequent growth in the upper atmosphere. The study finds motivation in previous observations that lightning strikes contribute much of the nitric acid observed in the upper atmosphere. The author's contribution is to demonstrate that nitric acid adds synergistically in the chamber, with the sulfuric acid and ammonia already present, to enhance rates of nanoparticle formation and growth to sufficient sizes needed to impact cloud droplet and ice particle formation. The author's results and presentation lead to improved understanding, quantification, and parameterization of the chemical mechanisms and reaction stoichiometry responsible for enhancement of these nucleation and growth processes.

Atmospheric implications are described in the final section of the article. The authors provide sound justification for their findings. If there is any weakness here, it is that little theoretical guidance is given to motivate and help the reader understand the use of power law fits to parameterize nucleation rates for use in atmospheric models. These fits are in fact rooted in the fundamental chemical law of mass action and on nucleation theorems for cluster stoichiometry derived from this law. The empirical log-log plot approach used in Fig. 2 for ammonia is effective, but the theoretical basis should also be (at least) mentioned. The suggestion in line 307 of “additional bonding between ammonia and nitric acid molecules in the nucleating clusters” is key to the paper and justifies its use of the word “synergistic” in the title. This is best understood by noting the fundamental connection between the empirical exponents and cluster stoichiometry nicely accounted for in the “nucleation theorems” of Kashchiev, Oxtoby, and others. It would take only an additional sentence or two to make this important connection.

In summary I find this paper to be interesting enough and important enough for publication in Nature.

Referee #2 (Remarks to the Author):

This study reports on new aerosol particle formation experiments carried out in the CLOUD chamber at CERN. The team investigated this process in mixtures of sulfuric acid, nitric acid and ammonia vapors at temperature and relative humidity conditions relevant for the upper troposphere. At elevated ammonia levels, new particles are observed to form as ternary H₂SO₄-NH₃-H₂O or (in Asian monsoon 'hotspot' conditions) quaternary HNO₃-H₂SO₄-NH₃-H₂O solution particles that may crystallize in suitable conditions.

The ice nucleation ability of the multi-acid particles was also investigated. It was found that growth of by uptake of ammonia and nitric acid generates INPs that form ice at relative humidity over ice as low as 120%. Backed up by earlier measurements, the data point to crystallization occurring at specific (rather low) sulfate/nitrate ratios as the cause of heterogeneous ice nucleation; the latter is described by empirical active surface site model.

In essence, this is what stated in the paragraph 239-270 and in the opening paragraph l15+16, both of which in my opinion are appropriate. However, the concluding paragraph states in l360+361 that the synergistic nucleation between the vapors 'is likely to be an important source of ... ice nuclei ...' which is a much stronger assertion that 'could provide an important source of ice nucleating particles ...' in l269+270.

Whether or not a particle identified as a good INP does not imply that it significantly alters cirrus cloud properties or that cirrus changes caused by INPs result in significant changes in cirrus radiative forcing. For instance, an INP can nucleate ice at low relative humidity, but if its number concentration at cirrus levels is low, it may not be able to compete in cirrus formation.

The authors have neither studied the impact of the HNO₃-H₂SO₄-NH₃-H₂O particles on the formation of (convective/in situ) cirrus, e.g., in cloud-resolving simulations, nor have they presented ice nucleation spectra and quantified ice active number concentrations (that may vary widely with strong hemispheric contrast). The multi-phase nature of the aerosol particles together with the complexity of real-world atmospheric temperature-humidity trajectories and the presence of other INPs make a sound assessment of the impact of the 'new' INP type on cirrus clouds extremely challenging. Only when these issues have been addressed, one might classify these INPs as truly important.

I understand that addressing these issues is not within the scope of this manuscript, which has a clear focus on new aerosol particle formation. I do think, though, that the ice nucleation aspect is a valuable addition that is worth being explored in greater detail in future work. In view of the above, my advice is to tone down the assertion in l269+270 regarding the ice nucleation aspect, which may turn out to be overstated.

Referee #3 (Remarks to the Author):

This paper uses careful experimental measurements to show that ammonia, sulphuric acid and nitric acid vapours nucleate to form particles more rapidly than any two of the vapours on their own at upper tropospheric temperatures. The presence of ammonium nitrate particles in Asian monsoon anticyclonic outflow in the UT and the mechanisms for their formation and dependence on surface ammonia emissions are topics of high current interest and importance. The importance and significance of the experimental finding in the current manuscript is framed in terms of the observations of ammonia by MIPAS in the UT in the Asian summer monsoon season and of solid ammonium nitrate by CRISTA (and supported by MIPAS).

As is the case for a large number of recent experimental findings from the CLOUD chamber, the measurements reported in the current paper are of high quality and have been conducted in well-designed and constructed experiments with state-of-the-science instrumentation and facilities. It is clear that nucleation of the three considered vapours is substantially enhanced over that in systems containing any two. I only have one question about the measurements. The IN ability measurements are welcome, though their relationship to ambient observations or the postulated mechanisms is a little unclear. It is also stated that "This finding is in agreement with previous studies with pure ammonium nitrate (NH_4NO_3) particles, showing that they exist as supercooled liquid solution droplets even at very low relative humidity". How does this fit with the reference 6 observations that are only consistent with solid ammonium nitrate, which is stated in this reference to form through glaciation even at high RH? The theoretical constructs in the current paper similarly appear appropriate for interpreting the measurements and deriving appropriate formulations of the nucleation parameterisation.

The main question I have for the authors is whether the sulphuric acid concentrations are supported by Asian Monsoon observations or whether the synergistic nucleation mechanism is conjecture based on possible / likely sulphuric acid concentrations. Clearly, sulphuric acid concentrations will be dependent on SA formation rate and available condensation sink. The latter condition requires the formation to be out-of-cloud and this needs to be in the presence of the nitric acid and ammonia vapours for the enhanced nucleation to take place. One of the postulated mechanisms for ammonia release is cloud droplet glaciation, which requires cloud. The mechanism for solid nitric acid postulated in reference 6 is the cooling of sulphate containing aqueous particles, even at high RH. How do the current findings fit with this interpretation? Is this consistent with processes included in the EMAC modelling?

I do not question the finding that the sulphuric acid substantially enhances ammonia-nitric acid nucleation and growth (and vice versa) and the plausibility of the conclusion, based on the experimental findings. I guess the question boils down to whether the "what if" plausible scenarios make a sufficiently compelling case for the importance of synergistic particle formation. Is it the only way to explain certain observations? If so, what are they? What are the previous postulations as discussed in the final paragraph of reference 6 and why don't they work? I think this needs to be drawn out a little better in the current manuscript to convincingly convey the importance of the studied process.

Referee #4 (Remarks to the Author):

This is a very interesting manuscript that shows that combining three nucleating agents together ($\text{HNO}_3/\text{H}_2\text{SO}_4/\text{NH}_3$) produces orders of magnitude higher nucleation rates than any combination of two components only. In addition, it shows that these newly nucleated particles can grow much faster in the same conditions to CCN sizes, from cases where only two components are considered. It is argued that this is relevant for the Asian monsoon conditions, especially at “hot spots” in the upper troposphere with enhanced pollution. The manuscript is very well written and deserves publication, but I do have my reservations on the applicability of the described processes in the global atmosphere. This applies to both new particle formation and ice nucleation, as described below.

New particle formation

It is not clear whether this process is relevant anywhere else outside the Asian monsoon region and time. Have the authors only focused on the Asian monsoon because this is the only place where upper tropospheric NH_3 is elevated? What happens to deep convection regions e.g. throughout the tropics? How about shallow convection, where both temperature and RH will be different from what was studied in the manuscript?

New particle formation was also measured in the upper troposphere over the Amazon basin, but it is not mentioned in the manuscript (<https://doi.org/10.5194/acp-18-921-2018>; reference 11 in the manuscript is a related one). In that study, the role of organic aerosols was key. How does that study affect the conclusions of the manuscript? There are plenty of organic aerosols in the Asian monsoon as well, can they dampen the role of the nucleation mechanism presented in the manuscript when one studies the real atmosphere? Also, how are the conclusions affected by the results from reference 11 in the Methods?

After the first particles are formed under the conditions studied, competition between further nucleation and condensation becomes significant. At which point nucleation becomes negligible due to the increased condensation sink of $\text{HNO}_3/\text{H}_2\text{SO}_4/\text{NH}_3$? The presence of organic vapors, especially at those low temperatures, can only complicate the picture.

Ice nucleation

It is argued that the newly formed particles (that grew to CCN and IN sizes) are as efficient ice nucleation agents as mineral dust. This is an overstatement, since a) only the highest sulfate/nitrate ratios approach the IN activity of mineral dust (figure 4a), and b) this does not happen to the fullest extent, since the y axis in figure 4a is log-scale. Statements like “nucleation followed by rapid growth [...] produces ice nucleating particles that are as efficient as typical desert dust particles at nucleating ice” need to be dialed down, if anything because the conditions needed are only demonstrated to occur at Asian monsoon hot spots only.

How would plot 4a look like if plotted against an increasing ratio of nitrate and then how for ammonium, instead of sulfate shown?

A description of how mixed-phase clouds are treated in the two models was not provided. This is key in understanding the impact those new particles have on ice cloud formation. Are these particles lost following freezing a droplet, or they remain available for further freezing? If the former, then they would have a much shorter lifetime than “from one week to one month in the upper troposphere”, which can reduce their global impact. If the latter, their impact on ice formation might be overly

exaggerated.

One can envision a third option: aerosols are lost in supercooled droplets, they freeze the droplets over, and then they eject NH_3 (as described in the manuscript is happening), becoming available to further nucleate particles, grow them, and freeze more clouds. It is stated in the manuscript that NH_3 is the limiting factor in the whole process, so regenerating NH_3 can act as a catalyst to new particle and cloud ice formation. In the absence of a description, this is only a speculation that happens in the models used.

RE: A point-by-point response to referee comments

We are grateful to the valuable comments from the anonymous referees and provide a point-by-point response to the comments below.

Referees' comments:

Referee #1

Review of “Synergistic particle formation in the upper troposphere by nitric acid, sulfuric acid and ammonia” by M. Wang et al.

The CLOUD chamber experiments, described by the authors, support nitric acid enhancement of sulfuric acid-ammonia in new particle formation. These is an interesting and important result that sheds new light on the multicomponent nucleation mechanisms controlling new particle formation and their subsequent growth in the upper atmosphere. The study finds motivation in previous observations that lightning strikes contribute much of the nitric acid observed in the upper atmosphere. The author’s contribution is to demonstrate that nitric acid adds synergistically in the chamber, with the sulfuric acid and ammonia already present, to enhance rates of nanoparticle formation and growth to sufficient sizes needed to impact cloud droplet and ice particle formation. The author’s results and presentation lead to improved understanding, quantification, and parameterization the chemical mechanisms and reaction stoichiometry responsible for enhancement of these nucleation and growth processes.

Atmospheric implications are described in the final section of the article. The authors provide sound justification for their findings. If there is any weakness here, it is that little theoretical guidance given to motivate and help the reader understand the use of power law fits to parameterize nucleation rates for use in atmospheric models. These fits are in fact rooted in the fundamental chemical law of mass action and on nucleation theorems for cluster stoichiometry derived from this law. The empirical log-log plot approach used in Fig. 2 for ammonia is effective, but the theoretical basis should also be (at least) mentioned. The suggestion in line 307 of “additional bonding between ammonia and nitric acid molecules in the nucleating clusters” is key to the paper and justifies its use of the word “synergistic” in the title. This is best understood by noting the fundamental connection between the empirical exponents and cluster stoichiometry nicely accounted for in the “nucleation theorems” of Kashchiev, Oxtoby, and others. It would take only an additional sentence or two to make this important connection.

In summary I find this paper to be interesting enough and important enough for publication in Nature.

Reply: We thank the referee for the positive comments. We have now added a sentence to establish the connection between the power law fits and the first nucleation theorem: “To evaluate its importance on a global scale, we first parameterised our experimentally measured $J_{1.7}$ for HNO_3 – H_2SO_4 – NH_3 nucleation as a function of sulfuric acid, nitric acid and ammonia concentrations (Methods). The parameterisation is obtained using a power law dependency for each vapour (Figure ED5), given that the critical cluster composition is associated with the exponents according to the first nucleation theorem (Oxtoby & Kashchiev, 1994).”.

Referee #2

This study reports on new aerosol particle formation experiments carried out in the CLOUD chamber at CERN. The team investigated this process in mixtures of sulfuric acid, nitric acid and ammonia vapors at temperature and relative humidity conditions relevant for the upper troposphere. At elevated ammonia levels, new particles are observed to form as ternary $H_2SO_4-NH_3-H_2O$ or (in Asian monsoon 'hotspot' conditions) quaternary $HNO_3-H_2SO_4-NH_3-H_2O$ solution particles that may crystallize in suitable conditions.

The ice nucleation ability of the multi-acid particles was also investigated. It was found that growth of by uptake of ammonia and nitric acid generates INPs that form ice at relative humidity over ice as low as 120%. Backed up by earlier measurements, the data point to crystallization occurring at specific (rather low) sulfate/nitrate ratios as the cause of heterogeneous ice nucleation; the latter is described by empirical active surface site model.

In essence, this is what stated in the paragraph 239-270 and in the opening paragraph 115+16, both of which in my opinion are appropriate. However, the concluding paragraph states in 1360+361 that the synergistic nucleation between the vapors 'is likely to be an important source of ... ice nuclei ...' which is a much stronger assertion that 'could provide an important source of ice nucleating particles ...' in 1269+270.

Whether or not a particle identified as a good INP does not imply that it significantly alters cirrus cloud properties or that cirrus changes caused by INPs result in significant changes in cirrus radiative forcing. For instance, an INP can nucleate ice at low relative humidity, but if its number concentration at cirrus levels is low, it may not be able to compete in cirrus formation.

The authors have neither studied the impact of the $HNO_3-H_2SO_4-NH_3-H_2O$ particles on the formation of (convective/in situ) cirrus, e.g. in cloud-resolving simulations, nor have they presented ice nucleation spectra and quantified ice active number concentrations (that may vary widely with strong hemispheric contrast). The multi-phase nature of the aerosol particles together with the complexity of real-world atmospheric temperature-humidity trajectories and the presence of other INPs make a sound assessment of the impact of the 'new' INP type on cirrus clouds extremely challenging. Only when these issues have been addressed, one might classify these INPs as truly important.

I understand that addressing these issues is not within the scope of this manuscript, which has a clear focus on new aerosol particle formation. I do think, though, that the ice nucleation aspect

is a valuable addition that is worth being explored in greater detail in future work. In view of the above, my advice is to tone down the assertion in l269+270 regarding the ice nucleation aspect, which may turn out to be overstated.

Reply: The referee's point is well taken. We have toned down our assertion in the last paragraph to "In summary, synergistic nucleation of nitric acid, sulfuric acid and ammonia could provide an important source of new cloud condensation nuclei and ice nuclei in the upper troposphere, especially over the Asian monsoon region, and is closely linked with anthropogenic ammonia emissions."

Referee #3

This paper uses careful experimental measurements to show that ammonia, sulphuric acid and nitric acid vapours nucleate to form particles more rapidly than any two of the vapours on their own at upper tropospheric temperatures. The presence of ammonium nitrate particles in Asian monsoon anticyclonic outflow in the UT and the mechanisms for their formation and dependence on surface ammonia emissions are topics of high current interest and importance. The importance and significance of the experimental finding in the current manuscript is framed in terms of the observations of ammonia by MIPAS in the UT in the Asian summer monsoon season and of solid ammonium nitrate by CRISTA (and supported by MIPAS).

As is the case for a large number of recent experimental findings from the CLOUD chamber, the measurements reported in the current paper are of high quality and have been conducted in well-designed and constructed experiments with state-of-the-science instrumentation and facilities. It is clear that nucleation of the three considered vapours is substantially enhanced over that in systems containing any two. I only have one question about the measurements. The IN ability measurements are welcome, though their relationship to ambient observations or the postulated mechanisms is a little unclear. It is also stated that "This finding is in agreement with previous studies with pure ammonium nitrate (NH₄NO₃) particles, showing that they exist as supercooled liquid solution droplets even at very low relative humidity". How does this fit with the reference 6 observations that are only consistent with solid ammonium nitrate, which is stated in this reference to form through glaciation even at high RH? The theoretical constructs in the current paper similarly appear appropriate for interpreting the measurements and deriving appropriate formulations of the nucleation parameterisation.

Reply: We appreciate the referee's question and have modified the manuscript to clarify this confusion. For the ice nucleation ability of ammonium nitrate particles, our experiments show that if particles presented as absolutely pure ammonium nitrate, they would exist as supercooled liquid droplets. The ambient observations of solid ammonium nitrate particles exactly demonstrate that these particles are not pure ammonium nitrate. The main emphasis of our measurements, however, is placed on the heterogeneous ice nucleation triggered by adding a trace amount of sulfate to the ammonium nitrate. Even in the very pure conditions of CLOUD we needed to take great care to accurately quantify sulfate well below 1 % by mass, which triggers efficient ice nucleation ability in the otherwise pure ammonium nitrate. This is consistent with the assertion in reference 6 that "Impurities of ammonium sulfate allow the crystallization of ammonium nitrate even in the conditions, such as a high relative humidity, that prevail in the upper troposphere." This agreement also highlights the existence and importance of all three components — nitric acid, sulfuric acid and ammonia — in the Asian monsoon upper troposphere.

The main question I have for the authors is whether the sulphuric acid concentrations are supported by Asian Monsoon observations or whether the synergistic nucleation mechanism is conjecture based on possible / likely sulphuric acid concentrations. Clearly, sulphuric acid concentrations will be dependent on SA formation rate and available condensation sink. The latter condition requires the formation to be out-of-cloud and this needs to be in the presence of the nitric acid and ammonia vapours for the enhanced nucleation to take place. One of the postulated mechanisms for ammonia release is cloud droplet glaciation, which requires cloud. The mechanism for solid nitric acid postulated in reference 6 is the cooling of sulphate containing aqueous particles, even at high RH. How do the current findings fit with this interpretation? Is this consistent with processes included in the EMAC modelling?

Reply: The referee makes a good point as sulfuric acid is one of the key components of this nucleation scheme. Aircraft measurements from the First Aerosol Characterization Experiment (ACE 1) ¹ and Pacific Exploratory Mission (PEM-Tropics A) ² have shown that sulfuric acid averages around 0.1 pptv ($\sim (0.63 - 2.5) \times 10^6 \text{ cm}^{-3}$) throughout most of the free troposphere. Sulfuric acid concentrations used in our experiments are thus consistent with the observations.

The referee is correct that both acids are formed by out-of-cloud oxidation while ammonia is possibly released during cloud glaciation. As such, it is likely that the synergistic particle formation occurs initially in the mixing zone between the cloud outflow and the background upper troposphere where the released ammonia mixes with pre-existing (background) sulfuric acid and nitric acid. Subsequently, as ammonia is titrated after several e-folding times or gradually diffuses away, this nucleation scheme will shift from ammonia-rich regime to ammonia-limited regime.

The EMAC simulation in this manuscript does not provide a sufficiently high spatial resolution to account for deep convection processes, which are instead parameterized. This could be the source of the difference between the EMAC and TOMCAT simulations. Therefore, additional ambient measurements and cloud resolving models, both with high spatial resolution, would be required to better resolve the source and formation mechanisms of tropospheric sulfuric acid and ammonia, and in turn to better constrain new particle formation processes in the upper troposphere. Nevertheless, current ambient measurements confirm the presence of ample ammonia, and our experiments show that synergistic $\text{HNO}_3\text{--H}_2\text{SO}_4\text{--NH}_3$ nucleation is a viable mechanism for new particle formation in the Asian monsoon upper troposphere.

I do not question the finding that the sulphuric acid substantially enhances ammonia-nitric acid nucleation and growth (and vice versa) and the plausibility of the conclusion, based on the experimental findings. I guess the question boils down to whether the “what if” plausible scenarios make a sufficiently compelling case for the importance of synergistic particle formation. Is it the only way to explain certain observations? If so, what are they? What are the previous postulations as discussed in the final paragraph of reference 6 and why don't they work? I think this needs to be drawn out a little better in the current manuscript to convincingly convey the importance of the studied process.

Reply: We are not sure what previous postulations the referee is referring to. Our reading of the final paragraph of reference 6 does not reveal any postulation on particle formation mechanisms. It reads:

“In the future, rising emissions of NH_3 will probably also lead to a change of AN particles in the UT with potential consequences for the Earth's radiative budget. For quantitative assessments using chemical–dynamical models, a better quantification of the NH_3 surface emissions over the Indian subcontinent as well as a deeper understanding of the interaction processes of NH_3 with liquid water and ice during convection is needed. Furthermore, it is most important to characterize the ice-nucleating capacity of solid AN particles.”

To the best of our knowledge, reference 6 has demonstrated that ground ammonia is a source of upper tropospheric ammonium nitrate particles, thus contributing to particle mass concentration. Here, our experiments, motivated by these observations, show that uplifted ammonia not only contribute to particle mass, but also substantially increase particle number. We also propose a novel synergistic particle formation mechanism to account for the increase.

Since there is almost no in-situ composition measurement of clusters and newly formed particles in the upper troposphere, we can only infer the major particle formation pathways from

indirect evidence such as chemical composition of precursor vapors or larger particles. Previously established mechanisms include binary and ternary sulfuric acid nucleation, which drive new particle formation over marine or anthropogenically influenced regions^{3–6}, as well as oxygenated organics nucleation, which dominates over pristine vegetated areas such as the Amazon basin^{7–9}.

The synergistic nucleation mechanism we propose appears to be an important pathway driving new particle formation in the Asian monsoon regions for at least two reasons: First, $\text{HNO}_3\text{--H}_2\text{SO}_4\text{--NH}_3$ nucleation is orders of magnitude faster than binary and ternary sulfuric acid nucleation, given the ammonia levels observed in the Asian monsoon upper troposphere; Second, the ammonium nitrate concentration is often higher than the sum of all other condensable vapors (presumably sulfuric acid and oxygenated organics), given that particle composition measurements in reference 6 have shown that ammonium nitrate can explain over half of the particulate volume.

Referee #4

This is a very interesting manuscript that shows that combining three nucleating agents together ($\text{HNO}_3/\text{H}_2\text{SO}_4/\text{NH}_3$) produces orders of magnitude higher nucleation rates than any combination of two components only. In addition, it shows that these newly nucleated particles can grow much faster in the same conditions to CCN sizes, from cases where only two components are considered. It is argued that this is relevant for the Asian monsoon conditions, especially at “hot spots” in the upper troposphere with enhanced pollution. The manuscript is very well written and deserves publication, but I do have my reservations on the applicability of the described processes in the global atmosphere. This applies to both new particle formation and ice nucleation, as described below.

New particle formation: It is not clear whether this process is relevant anywhere else outside the Asian monsoon region and time. Have the authors only focused on the Asian monsoon because this is the only place where upper tropospheric NH_3 is elevated? What happens to deep convection regions e.g. throughout the tropics?

Reply: We have focused on the Asian monsoon region in part because ammonia concentrations measured in this region are by far the highest in the upper troposphere, but also because this region is fairly extensive. While we frame this synergistic nucleation in a scenario that suits the Asian monsoon region, the physics applies globally — the colder the conditions are, the more important this mechanism is likely to be.

Given the typical acid-excess conditions in the upper troposphere, the only constraint is

availability of ammonia, which is not yet well constrained. While satellite-based ammonia measurements have provided a spatial distribution on a global scale, they are limited to cloud-free areas due to blockage of the ammonia signal by optically thick clouds. However, we propose that cloud glaciation may well be a major pathway for ammonia vapor release into the upper troposphere; this process then may not be captured by satellites given that it is by definition near clouds, along with the short life-time and high spatial heterogeneity of gas-phase ammonia. This may also explain why the in-situ measured ammonia concentrations are up to 40 times higher than those from satellite measurements¹⁰. Therefore, additional measurements of the ammonia spatial distribution via in-situ measurements is needed in order to assess the role of this synergistic nucleation in other deep convection regions. Nevertheless, as global ammonia emissions continue to increase due to agricultural growth and the warmer climate^{11,12}, the importance of this synergistic nucleation mechanism is likely to increase.

How about shallow convection, where both temperature and RH will be different from what was studied in the manuscript?

Reply: The measurements we present here are confined to the very cold conditions of the upper troposphere – this is due to the highly constrained scheduling of CLOUD experiments and the extensive phase space even at one temperature. For this reason we focus our manuscript on these conditions. However, we can speculate on higher temperature conditions based on some constraints. After considering the following key factors, we surmise that synergistic HNO₃–H₂SO₄–NH₃ nucleation may still be an important particle formation pathway during shallow convection:

Temperature dependence: In a previous study¹³ we showed that pure HNO₃–NH₃ nucleation occurs below about 258 K, though it is relatively slow. Synergistic HNO₃–H₂SO₄–NH₃ nucleation thus should occur around or slightly above 258 K. This corresponds to ambient temperature at a few kilometers altitude in the free troposphere, where shallow convection can often reach. Therefore, temperature at the outflow of shallow convection should favor this synergistic nucleation scheme.

Ammonia availability: Current understanding is that ammonia keeps being scavenged while being entrained into the free troposphere during shallow convection. Ammonia availability is thus a question of timing and source concentrations, which often vary from region to region. However, up to ppbv levels of ammonia have been observed in the free troposphere reaching altitudes of about 6 km during in-situ aircraft measurements^{14,15}. Therefore, ammonia may well be sufficient for synergistic HNO₃–H₂SO₄–NH₃ nucleation to take place. However, more detailed information about ammonia vertical profile and more laboratory experiments are needed to map out the boundary conditions for this nucleation scheme.

Relative humidity (RH): On one hand, sulfuric-acid-driven particle formation in the free tropo-

sphere has been observed when RH is elevated ³, possibly due to cluster stabilization by water molecules via either forming additional hydrogen bonding with acid and base, or evaporating to take away excess energy. On the other hand, new particle formation events have been observed preferentially at low RH in pristine and anthropogenic environments, possibly because high RH is often associated with low solar radiation intensity and a high condensation sink ¹⁶. Nonetheless, global model simulations ¹⁷ have shown that RH changes only influence tropospheric 3-nm particle concentrations marginally (less than 15 %), especially at altitudes where variation in water concentration is way less than in concentrations of sulfuric acid and ammonia.

With all said and done, we decided to confine our focus to the upper tropospheric conditions directly addressed by our experiments, which is also where ambient observations indicate that this mechanism may well be important.

New particle formation was also measured in the upper troposphere over the Amazon basin, but it is not mentioned in the manuscript (<https://doi.org/10.5194/acp-18-921-2018>; reference 11 in the manuscript is a related one).

Reply: This is an important study, we have now cited it in our main text.

In that study, the role of organic aerosols was key. How does that study affect the conclusions of the manuscript? There are plenty of organic aerosols in the Asian monsoon as well, can they dampen the role of the nucleation mechanism presented in the manuscript when one studies the real atmosphere? Also, how are the conclusions affected by the results from reference 11 in the Methods? After the first particles are formed under the conditions studied, competition between further nucleation and condensation becomes significant. At which point nucleation becomes negligible due to the increased condensation sink of HNO₃/H₂SO₄/NH₃? The presence of organic vapors, especially at those low temperatures, can only complicate the picture.

Reply: We agree with the referee that the presence of organic vapors can complicate the picture. It is still unclear whether inorganic (acid-base) and organic particle formation processes are synergistic, parallel, or even antagonistic. At this stage, we can only infer the major particle formation pathways from indirect evidence such as chemical composition of precursor vapors or larger particles. For example, over pristine vegetated areas such as the Amazon basin, composition measurements indicate particle formation may be dominated by organic vapors ^{7,9}. Over agricultural areas such as the Asian monsoon regions, however, ammonium nitrate can often explain over half of the particulate volume ¹⁰. This means ammonium nitrate concentration is higher than the sum of all other condensable vapors (presumably sulfuric acid and oxygenated organics). We therefore can infer that synergistic HNO₃–H₂SO₄–NH₃ nucleation is a major particle formation pathway. It

seems unlikely that this inorganic pathway and the organic pathways are antagonistic in growth, and without strong indications otherwise it seems likely they are more or less additive for nucleation itself. However, to further investigate interactions between different nucleation schemes we would rely on additional information on the source and identity of organic vapors that are present in the Asian monsoon upper troposphere.

As for the competition between further nucleation and condensation, condensation will always win because the energy barrier is much higher for vapor molecules to nucleate than it is to condense. This is why nucleation would never occur under equilibrium conditions. However, there are ample sources of inhomogeneity in the ambient atmosphere: photo-oxidation producing nitric acid and sulfuric acid is a good example, and deep convection uplifting ammonia into upper troposphere is another. It is thus likely that the outflow conditions in the Asian monsoon regions will typically include strong inhomogeneities maintaining supersaturation of nitric acid, sulfuric acid and ammonia with sufficient magnitude to drive particle formation. After that, this particle formation process will persist until ammonia is depleted after several e-folding times set by the particle condensation sink. This time scale will be several hours, based on condensation sinks generally observed in the tropical upper troposphere⁶. Condensation sink also regulates the apparent particle formation rate, but it is particle survival probability and not nucleation rate per se that is suppressed by the increased condensation sink. Particle nucleation and survival should be considered separately because changes in particle survival probability from the competition between condensation growth and coagulation loss do not exert influence on either nucleation mechanisms or their relative importance.

Ice nucleation: It is argued that the newly formed particles (that grew to CCN and IN sizes) are as efficient ice nucleation agents as mineral dust. This is an overstatement, since a) only the highest sulfate/nitrate ratios approach the IN activity of mineral dust (figure 4a), and b) this does not happen to the fullest extent, since the y axis in figure 4a is log-scale. Statements like “nucleation followed by rapid growth [...] produces ice nucleating particles that are as efficient as typical desert dust particles at nucleating ice” need to be dialed down, if anything because the conditions needed are only demonstrated to occur at Asian monsoon hot spots only.

Reply: The referee’s point is well taken. We have now dialed down the sentence to “Our measurements show that $\text{HNO}_3\text{--H}_2\text{SO}_4\text{--NH}_3$ nucleation followed by rapid growth from nitric acid and ammonia condensation—which results in low sulfate/nitrate—produces ice nucleating particles that are comparable to typical desert dust particles at nucleating ice”.

How would plot 4a look like if plotted against an increasing ratio of nitrate and then how for

ammonium, instead of sulfate shown?

Reply: While we did not conduct such measurements as the referee suggested, we can infer the ice nucleation behaviors from previous results. This question boils down to a comparison between the ice nucleation abilities of ammonium nitrate–ammonium sulfate particles, ammonium sulfate particles, and acidic sulfate particles. Because the crystallized ammonium nitrate particles have similar ice nucleation ability as dry ammonium sulfate particles at cirrus temperatures^{18,19}, adding (ammonium) nitrate to ammonium sulfate particles may not alter the overall ice nucleation active surface site density of the crystalline particles. Moreover, because fully-neutralized ammonium sulfate particles are more efficient ice nuclei than acidic sulfate particles²⁰, adding ammonia to sulfuric acid(–nitric acid) particles would promote the freezing via transitioning from homogeneous ice nucleation to heterogeneous ice nucleation.

A description of how mixed-phase clouds are treated in the two models was not provided. This is key in understanding the impact those new particles have on ice cloud formation. Are these particles lost following freezing a droplet, or they remain available for further freezing? If the former, then they would have a much shorter lifetime than “from one week to one month in the upper troposphere”, which can reduce their global impact. If the latter, their impact on ice formation might be overly exaggerated. One can envision a third option: aerosols are lost in supercooled droplets, they freeze the droplets over, and then they eject NH₃ (as described in the manuscript is happening), becoming available to further nucleate particles, grow them, and freeze more clouds. It is stated in the manuscript that NH₃ is the limiting factor in the whole process, so regenerating NH₃ can act as a catalyst to new particle and cloud ice formation. In the absence of a description, this is only a speculation that happens in the models used.

Reply: We agree that mixed-phase processes are likely to be crucial; like all cloud processes they are heavily parameterized in the global scale models. We feel that delving into the specifics is beyond the scope of this work – our focus is on the CLOUD experimental observations motivated by ambient measurements that confirm the likely importance of this mechanism in the upper troposphere, at least in the Asian Monsoon region. However, it is not obvious to us that the impact of either the HNO₃–H₂SO₄–NH₃ nucleation or the ice nucleating capacity of the resulting particles would be overly exaggerated without reinvigoration by cyclic NH₃ release (though this is a fascinating suggestion). First, the number concentration of Aitken-mode particles in the tropical upper troposphere is of $\gtrsim 100 \text{ cm}^{-3}$ ⁶, while ice crystal concentrations in clouds are $\lesssim 0.1 \text{ cm}^{-3}$ ²¹. Generally, only a small fraction of ice nucleating particles actually become ice crystals. Therefore, the lifetimes of nucleation-mode and Aitken-mode particles are likely still determined by coagulation;

being good ice nucleating particles does not change this fact. Second, as the referee indicates, particles remaining available for further freezing would enhance their impact on ice formation, but we emphasize that our mechanism envisions particle formation (and crystallization) outside of clouds. It is confirmed by our measurements that ammonium nitrate–ammonium sulfate particles are good ice nuclei. That these particles spread across extensive areas instead of nucleating ice all at once is most likely due to limited water supersaturation in the upper troposphere.

As for aerosol-cloud interactions, ice nucleating particles are often not required for ice formation in the upper troposphere as water freezes homogeneously below 238 K, and thus particles hardly interact with clouds. Ice nuclei and cloud condensation nuclei are important, however, as they descend to around 400 and 900 hPa, where mixed phase clouds are sensitive to their abundance. Then after activation, particles are either removed by precipitation or released to the atmosphere by evaporation. But since the mean descent rate is less than 15 hPa/day for the tropical upper troposphere ⁶ (10 days from 250 to 400 hPa), particle lifetimes will not be influenced much by this process either.

As for ammonia re-evaporation, droplet crystallization will drive the dissolved ammonia molecules and ammonium ions to the thin air-liquid interface, due to their increased activity coefficients in the crystal phase. Consequently, condensed-phase activities of ammonia and ammonium will become substantially larger, so will be the ammonia vapor pressure over the air-liquid interface. However, molecular dynamic simulations demonstrate that the thin air-liquid layer can often accommodate 94 – 100 % ammonia molecules ²². This means in stable cloud conditions, ammonia in the air-liquid interface of the ice particles will not evaporate easily, consistent with the near-unity retention coefficient measured in laboratory ²³. However, this air-liquid interface is unlikely to remain in deep convective clouds ^{22,24}, ammonia are thus highly mobile at the interface and prone to evaporation upon collisions of ice particles during deep convection. Therefore, it is uncertain that particle dissolution followed by cloud glaciation will still be a major source of ammonia outside deep convection. But we agree with the referee that this may be an interesting process for future experimental investigation.

We can address the model parameterizations, but delving deeply into them for this work would be a distraction, in our opinion. For more details on mixed phase cloud in the EMAC model, we kindly refer the referee to previous publications ^{25,26}. Moreover, since TOMCAT is a chemical transport model, it does not predict the impact of the new particles on ice formation.

References

1. Bates, T. S., Huebert, B. J., Gras, J. L., Griffiths, F. B. & Durkee, P. A. International global atmospheric chemistry (IGAC) project's first aerosol characterization experiment (ACE 1): Overview. *Journal of Geophysical Research: Atmospheres* **103**, 16297–16318 (1998).
2. Hoell, J. M. *et al.* Pacific Exploratory Mission in the tropical Pacific: PEM-Tropics A, August-September 1996. *Journal of Geophysical Research: Atmospheres* **104**, 5567–5583 (1999).
3. Clarke, A. *et al.* Nucleation in the equatorial free troposphere: Favorable environments during PEM-Tropics. *Journal of Geophysical Research: Atmospheres* **104**, 5735–5744 (1999).
4. Twohy, C. H. *et al.* Deep convection as a source of new particles in the midlatitude upper troposphere. *Journal of Geophysical Research: Atmospheres* **107**, AAC 6–1–AAC 6–10 (2002).
5. Lee, S.-H. *et al.* Particle formation by ion nucleation in the upper troposphere and lower stratosphere. *Science* **301**, 1886–1889 (2003).
6. Williamson, C. J. *et al.* A large source of cloud condensation nuclei from new particle formation in the tropics. *Nature* **574**, 399–403 (2019).
7. Weigel, R. *et al.* In situ observations of new particle formation in the tropical upper troposphere: the role of clouds and the nucleation mechanism. *Atmospheric Chemistry and Physics* **11**, 9983–10010 (2011).
8. Waddicor, D. A. *et al.* Aerosol observations and growth rates downwind of the anvil of a deep tropical thunderstorm. *Atmospheric Chemistry and Physics* **12**, 6157–6172 (2012).
9. Andreae, M. O. *et al.* Aerosol characteristics and particle production in the upper troposphere over the Amazon Basin. *Atmospheric Chemistry and Physics* **18**, 921–961 (2018).
10. Höpfner, M. *et al.* Ammonium nitrate particles formed in upper troposphere from ground ammonia sources during Asian monsoons. *Nature Geoscience* **12**, 608–612 (2019).
11. Erisman, J. W., Sutton, M. A., Galloway, J., Klimont, Z. & Winiwarter, W. How a century of ammonia synthesis changed the world. *Nature Geoscience* **1**, 636–639 (2008).
12. Warner, J. X. *et al.* Increased atmospheric ammonia over the world's major agricultural areas detected from space. *Geophysical Research Letters* **44**, 2875–2884 (2017).
13. Wang, M. *et al.* Rapid growth of atmospheric nanoparticles by nitric acid and ammonia condensation. *Nature* **580**, 184–189 (2020).
14. Schiferl, L. D. *et al.* Interannual variability of ammonia concentrations over the united states: sources and implications. *Atmospheric Chemistry and Physics* **16**, 12305–12328 (2016).
15. Guo, X. *et al.* Validation of IASI satellite ammonia observations at the pixel scale using in situ vertical profiles. *Journal of Geophysical Research: Atmospheres* **126**, e2020JD033475 (2021).
16. Kerminen, V.-M. *et al.* Atmospheric new particle formation and growth: review of field observations. *Environmental Research Letters* **13**, 103003 (2018).
17. Dunne, E. M. *et al.* Global atmospheric particle formation from CERN CLOUD measurements. *Science* **354**, 1119–1124 (2016).

18. Shilling, J. E., Fortin, T. J. & Tolbert, M. A. Depositional ice nucleation on crystalline organic and inorganic solids. *Journal of Geophysical Research: Atmospheres* **111** (2006).
19. Wagner, R. *et al.* Solid ammonium nitrate aerosols as efficient ice nucleating particles at cirrus temperatures. *Journal of Geophysical Research: Atmospheres* **125**, e2019JD032248 (2020).
20. Abbatt, J. *et al.* Solid ammonium sulfate aerosols as ice nuclei: A pathway for cirrus cloud formation. *Science* **313**, 1770–1773 (2006).
21. Gryspeerdt, E. *et al.* Ice crystal number concentration estimates from lidar–radar satellite remote sensing – Part 2: Controls on the ice crystal number concentration. *Atmospheric Chemistry and Physics* **18**, 14351–14370 (2018).
22. Ge, C., Zhu, C., Francisco, J. S., Zeng, X. C. & Wang, J. A molecular perspective for global modeling of upper atmospheric NH₃ from freezing clouds. *Proceedings of the National Academy of Sciences* **115**, 6147–6152 (2018).
23. Blohn, N. *et al.* The retention of ammonia and sulfur dioxide during riming of ice particles and dendritic snow flakes: laboratory experiments in the Mainz vertical wind tunnel. *Journal of Atmospheric Chemistry* **70** (2013).
24. Mari, C., Jacob, D. J. & Bechtold, P. Transport and scavenging of soluble gases in a deep convective cloud. *Journal of Geophysical Research: Atmospheres* **105**, 22255–22267 (2000).
25. Bacer, S. *et al.* Implementation of a comprehensive ice crystal formation parameterization for cirrus and mixed-phase clouds in the emac model (based on messy 2.53). *Geoscientific Model Development* **11**, 4021–4041 (2018).
26. Bacer, S. *et al.* Cold cloud microphysical process rates in a global chemistry–climate model. *Atmospheric Chemistry and Physics* **21**, 1485–1505 (2021).

Reviewer Reports on the First Revision:

Referees' comments:

Referee #1 (Remarks to the Author):

The authors have addressed my principal concern

Referee #2: No further comments addressed to you

Referee #3 (Remarks to the Author):

This review relates to the authors' response to my original "reviewer 3" concerns to the manuscript.

The authors have addressed my first comment on how their observations are "...in agreement with previous studies with pure ammonium nitrate (NH_4NO_3) particles, showing that they exist as supercooled liquid solution droplets even at very low relative humidity" adequately in the clarification in the rebuttal and manuscript that high RH crystallisation is caused by "impurities of ammonium sulphate" consistent with the referenced paper.

My second comment is largely addressed by the statement in the rebuttal that "it is likely that the synergistic particle formation occurs initially in the mixing zone between the cloud outflow and the background upper troposphere where the released ammonia mixes with pre-existing (background) sulfuric acid and nitric acid. Subsequently, as ammonia is titrated after several e-folding times or gradually diffuse away, this nucleation scheme will shift from ammonia-rich regime to ammonia-limited regime." It is then further acknowledged that the EMAC simulations are at insufficient resolution to capture such processes and that further model exploration should be conducted. It is further contested that the synergistic nucleation is a viable nucleation mechanism. Again, I agree that it is plausible, though firm support for its role is still tantalisingly evasive.

Finally, the authors attempt to address whether their synergistic nucleation mechanism is an important pathway by making comparisons between its rate and those of other established mechanisms and by a comparative statement about the ammonium nitrate concentration. This still appears short of the establishment of whether the mechanism is the only way to explain certain observations, rather than a plausible way to explain observed fractional ammonium nitrate volume.

I do not doubt the rapidity of the nucleation mechanism, nor the plausibility of its importance. On balance, I think that firm support for its role is not possible, nor should it be expected, at the current state-of-the-science. The manuscript is probably a sufficient demonstration of the plausibility of a newly postulated mechanism, notwithstanding its dependence on availability of a highly uncertain ammonia burden (prompting guidance for improved measurement coverage).

Referee #4 (Remarks to the Author):

I would like to thank the authors for the extremely detailed answers they provided to my comments. It is truly rare to get so detailed replies, and I applaud the authors for their time and work on these. I am satisfied with their replies in general, although I was expecting a few more details in the mixed phase clouds question in models (beyond the manuscript text, so it wouldn't had been a "distraction").

I am ready to recommend the manuscript for publication, but do have one request: I would expect that some of the questions I raised will be questions to future readers too. It is my understanding that hardly anything from the authors' replies made it into the manuscript. I do understand that there are space constraints, but some answers contain very valuable information that it would be a shame to get lost in a "reply to reviewers" document that nobody will read again. It would be great if some of the key statements make it into the text, or the methods. I leave to the discretion of the authors to decide which, based on their expert judgement.

Author Rebuttals to First Revision:

Referees' comments:

Referee #1

The authors have addressed my principal concern.

Reply: **We thank the referee for the positive comment.**

Referee #2

No further comments addressed to you.

Referee #3

This review relates to the authors' response to my original "reviewer 3" concerns to the manuscript. The authors have addressed my first comment on how their observations are "...in agreement with previous studies with pure ammonium nitrate (NH₄NO₃) particles, showing that they exist as supercooled liquid solution droplets even at very low relative humidity" adequately in the clarification in the rebuttal and manuscript that high RH crystallisation is caused by "impurities of ammonium sulphate" consistent with the referenced paper.

My second comment is largely addressed by the statement in the rebuttal that "it is likely that the synergistic particle formation occurs initially in the mixing zone between the cloud outflow and the background upper troposphere where the released ammonia mixes with pre-existing (background) sulfuric acid and nitric acid. Subsequently, as ammonia is titrated after several e-folding times or gradually diffuse away, this nucleation scheme will shift from ammonia-rich regime to ammonia-limited regime." It is then further acknowledged that the EMAC simulations are at insufficient resolution to capture such processes and that further model exploration should be conducted. It is further contested that the synergistic nucleation is a viable nucleation mechanism. Again, I agree that it is plausible, though firm support for its role is still tantalisingly evasive.

Finally, the authors attempt to address whether their synergistic nucleation mechanism is an important pathway by making comparisons between its rate and those of other established mechanisms and by a comparative statement about the ammonium nitrate concentration. This still appears short of the establishment of whether the mechanism is the only way to explain certain observations, rather than a plausible way to explain observed fractional ammonium nitrate volume.

I do not doubt the rapidity of the nucleation mechanism, nor the plausibility of its importance.

On balance, I think that firm support for its role is not possible, nor should it be expected, at the current state-of-the-science. The manuscript is probably a sufficient demonstration of the plausibility of a newly postulated mechanism, notwithstanding its dependence on availability of a highly uncertain ammonia burden (prompting guidance for improved measurement coverage).

Reply: We appreciate the referee's constructive comments. The careful work of the referee provides important feedback. We have now added a subsection on atmospheric interpretation of our data in the Methods section, to incorporate the key information discussed during the review process.

Referee #4

I would like to thank the authors for the extremely detailed answers they provided to my comments. It is truly rare to get so detailed replies, and I applaud the authors for their time and work on these. I am satisfied with their replies in general, although I was expecting a few more details in the mixed phase clouds question in models (beyond the manuscript text, so it wouldn't had been a "distraction").

I am ready to recommend the manuscript for publication, but do have one request: I would expect that some of the questions I raised will be questions to future readers too. It is my understanding that hardly anything from the authors' replies made it into the manuscript. I do understand that there are space constraints, but some answers contain very valuable information that it would be a shame to get lost in a "reply to reviewers" document that nobody will read again. It would be great if some of the key statements make it into the text, or the methods. I leave to the discretion of the authors to decide which, based on their expert judgement.

Reply: We thank the referee for the thorough review. As the referee suggested, we have now added a subsection on atmospheric interpretation of our data in the Methods section, to incorporate the key information discussed during the review process.